# Compositional Planning with Jumpy World Models

**Jesse Farebrother** [1 2 *]   **Matteo Pirotta** [3]   **Andrea Tirinzoni** [3]   **Marc G. Bellemare** [1 2 †]   **Alessandro Lazaric** [3]
**Ahmed Touati** [3]

## Abstract

The ability to plan with temporal abstractions is central to intelligent decision-making. Rather than reasoning over primitive actions, we study agents that compose pre-trained policies as temporally extended actions, enabling solutions to complex tasks that no constituent alone can solve. Such compositional planning remains elusive as compounding errors in long-horizon predictions make it challenging to estimate the visitation distribution induced by sequencing policies. Motivated by the *geometric policy composition* framework introduced in Thakoor et al. (2022), we address these challenges by learning predictive models of multi-step dynamics — so-called *jumpy world models* — that capture state occupancies induced by pre-trained policies across multiple timescales in an off-policy manner. Building on Temporal Difference Flows (Farebrother et al., 2025), we enhance these models with a novel consistency objective that aligns predictions across timescales, improving long-horizon predictive accuracy. We further demonstrate how to combine these generative predictions to estimate the value of executing arbitrary sequences of policies over varying timescales. Empirically, we find that compositional planning with jumpy world models significantly improves zero-shot performance across a wide range of base policies on challenging manipulation and navigation tasks, yielding, on average, a 200% relative improvement over planning with primitive actions on long-horizon tasks.

## 1. Introduction

In recent years, the success of large-scale foundation models in domains such as computer vision (e.g., Radford et al., 2021; Ravi et al., 2025; Assran et al., 2025) and natural language processing (e.g., Meta, 2024; OpenAI, 2024; Google DeepMind, 2025) has inspired a similar shift in Reinforcement Learning (RL). Foundation policies pre-trained on diverse unlabeled data or via unsupervised objectives can now generalize to a wide range of downstream tasks, without additional training or explicit planning. This has led to remarkable progress in areas like humanoid control (e.g., Peng et al., 2021; 2022; Tessler et al., 2023; 2024; Tirinzoni et al., 2025; Alegre et al., 2025) and real-world robotics (e.g., Brohan et al., 2023b;a; Luo et al., 2024; Ghosh et al., 2024; Black et al., 2024; 2025; NVIDIA, 2025; Li et al., 2026).

Despite these advances, a key limitation persists: while foundation policies can handle many tasks out of the box, they often fall short when faced with complex, long-horizon problems that require reasoning over extended sequences of decisions. In such cases, the planning horizon of a single policy is insufficient, and agents must compose policies to achieve their goals (Schmidhuber, 1991; Singh, 1992; Dayan & Hinton, 1992; Kaelbling, 1993b;a; Parr & Russell, 1997; Dietterich, 1998; Sutton et al., 1999; Precup, 2000).

Hierarchical RL (Barto & Mahadevan, 2003; Klissarov et al., 2025), including the options framework (Sutton et al., 1999), aims to achieve compositionality by training task-specific high-level policies to leverage manual (e.g., Nachum et al., 2018; Barreto et al., 2019; Carvalho et al., 2023; Park et al., 2023; 2025b) or automatic (e.g., Bacon et al., 2017; Machado et al., 2017; 2018; 2023; Bagaria et al., 2021; Sutton et al., 2023) task decompositions. In this paper, we take a fundamentally different approach: instead of learning task-specific hierarchies, we develop a framework for direct compositional planning over parameterized policies, requiring *no task-specific* training. By learning "jumpy" multi-step dynamics models – also known as "jumpy world models" (Murphy, 2024) – we enable flexible composition of existing policies, transforming how agents tackle novel tasks through intelligent recombination of existing behavior.

To operationalize this idea, we propose learning a *policy* and *horizon* conditioned *jumpy world model* that captures the distribution of future states for all parameterized policies over a continuum of geometrically decaying time horizons (Janner et al., 2020; Thakoor et al., 2022). To make

---

*Work done at Meta †CIFAR AI Chair [1] McGill University [2] Mila - Québec AI Institute [3] FAIR at Meta. Correspondence to: Jesse Farebrother <jfarebro@cs.mcgill.ca>.

this possible, we first generalize the recent Temporal Difference Flow (Farebrother et al., 2025) framework using a novel consistency objective that enforces coherence between predictions at different timescales, consistently improving long-horizon predictions. Additionally, we develop a novel estimator of the value of executing an arbitrary sequence of policies, each with its own variable timescale. With these two pieces in place, we demonstrate how to plan over a wide range of parameterized policies, allowing us to flexibly compose behaviors to solve complex, long-horizon tasks without the need for further environment interaction or fine-tuning. Empirically, across multiple classes of base policies evaluated on a suite of OGBench navigation and manipulation tasks (Park et al., 2025a), behavior-level planning consistently improves over zero-shot performance, often by a large margin. Finally, our approach outperforms state-of-the-art hierarchical baselines as well as alternative planning methods; in particular, planning with jumpy world models achieves a 200% relative improvement over action-level planning with a one-step world model on long-horizon tasks. These results demonstrate that planning with jumpy world models offers a powerful complement that is particularly effective for long-horizon decision-making.

## 2. Background

We model the environment as a reward-free discounted Markov Decision Process (MDP) defined as the 4-tuple $\mathcal{M} = (\mathsf{S}, \mathsf{A}, P, \gamma)$. Here, $\mathsf{S}$ and $\mathsf{A}$ represent the state and action spaces, respectively; $P : \mathsf{S} \times \mathsf{A} \to \mathscr{P}(\mathsf{S})$ characterizes the distribution over next states; and $\gamma \in [0, 1)$ is the discount factor. At each step $k$, the agent follows a policy $\pi : \mathsf{S} \to \mathscr{P}(\mathsf{A})$, generating a trajectory of state-action pairs $(S_k, A_k)_{k \geq 0}$ where $A_k \sim \pi(\cdot \mid S_k)$ and $S_k \sim P(\cdot \mid S_{k-1}, A_{k-1})$. When unambiguous, we use $S'$ as the immediate next state of $S$, and $S^+$ for a successor state at some future step $k > 0$. We use $\Pr(\cdot \mid S_0 = s, A_0 = a, \pi)$ and $\mathbb{E}[\cdot \mid S_0 = s, A_0 = a, \pi]$ to denote the probability and expectation over sequences induced by starting from $(s, a)$ and following $\pi$ thereafter.

**Successor Measure** For a policy $\pi$ and an initial state-action pair $(s, a)$, *the (normalized) successor measure* (Dayan, 1993; Blier et al., 2021), denoted by $m_\gamma^\pi(\cdot \mid s, a)$, is a probability measure over the state space $\mathsf{S}$. For any subset $\mathsf{X} \subseteq \mathsf{S}$, it represents the cumulative discounted probability of visiting a state in $\mathsf{X}$, discounted geometrically according to the time of arrival. Formally, this is defined as:

$$m_\gamma^\pi(\mathsf{X} \mid s, a) = (1 - \gamma) \tag{1}$$
$$\times \sum_{k=0}^\infty \gamma^k \Pr(S_{k+1} \in \mathsf{X} \mid S_0 = s, A_0 = a, \pi).$$

The normalization factor $1 - \gamma$ ensures $m_\gamma^\pi$ is a probability distribution. This admits an intuitive interpretation: rather than viewing $\gamma$ as a discount factor, one may equivalently consider an *auxiliary* process with a geometrically distributed lifetime – halting at each step with probability $1 - \gamma$ (Derman, 1970). Under this view, $m_\gamma^\pi(\mathsf{X} \mid s, a)$ equivalently characterizes the probability that the state visited at the halting time lies in $\mathsf{X}$. This yields a convenient reparameterization of the action-value function as:

$$Q_\gamma^\pi(s, a) = \mathbb{E}\Big[ \sum_{k=0}^\infty \gamma^k r(S_{k+1}) \mid S_0 = s, A_0 = a, \pi \Big]$$
$$\equiv (1 - \gamma)^{-1} \mathbb{E}_{S^+ \sim m_\gamma^\pi(\cdot \mid s, a)} \big[ r(S^+) \big], \tag{2}$$

expressing value as the expected reward at the geometrically distributed halting time scaled by the average lifetime $(1 - \gamma)^{-1}$. Note that this equivalence is purely mathematical[1]; the underlying MDP remains unchanged.

**Geometric Horizon Model** A Geometric Horizon Model (GHM; Janner et al., 2020; Thakoor et al., 2022) instantiates a jumpy world model as a *generative model* of the successor measure. It can be learned off-policy via temporal-difference learning, exploiting the fact that $m_\gamma^\pi$ is a fixed point of the Bellman equation:

$$m_\gamma^\pi(\cdot \mid s, a) = (1 - \gamma) P(\cdot \mid s, a) + \gamma \mathop{\mathbb{E}}_{\substack{S' \sim P(\cdot \mid s, a) \\ A' \sim \pi(\cdot \mid s)}} \big[ m_\gamma^\pi(\cdot \mid S', A') \big]. \tag{3}$$

Due to the Bellman equation's reliance on bootstrapping, the choice of generative model is critical for maintaining stability. Farebrother et al. (2025) demonstrate that prior approaches suffer from systemic bias at long horizons due to these bootstrapped predictions. To address this, Farebrother et al. (2025) propose the use of flow matching techniques (Lipman et al., 2023; Albergo & Vanden-Eijnden, 2023), which construct probability paths that evolve smoothly from a source distribution to the desired target distribution. By designing these paths to exploit structure in the temporal difference target, they show that bootstrapping bias can be controlled, enabling accurate long-horizon predictions.

In this framework, the GHM models a $d$-dimensional continuous state space[2] as an ordinary differential equation (ODE) parameterized by a time-dependent vector field $v_t : \mathbb{R}^d \times \mathsf{S} \times \mathsf{A} \to \mathbb{R}^d$. Sampling from the GHM begins by drawing initial noise $X_0 \in \mathbb{R}^d$ from a prior distribution $p_0 \in \mathscr{P}(\mathbb{R}^d)$ and subsequently following the flow

---

[1]This preserves expected cumulants and occupancy measures, but not trajectory-level statistics (Bellemare et al., 2023)

[2]While we assume a continuous state space $\mathsf{S} \subseteq \mathbb{R}^d$ for ease of exposition, flow matching readily extends to non-Euclidean and discrete spaces (e.g., Huang et al., 2022; Chen & Lipman, 2024; Gat et al., 2024; Kapusniak et al., 2024).

$\psi_t : \mathbb{R}^d \times \mathsf{S} \times \mathsf{A} \to \mathbb{R}^d$, defined by the following Initial Value Problem (IVP) for $t \in [0,1]$:

$$\frac{\mathrm{d}}{\mathrm{d}t} \psi_t(x \mid s, a) = v_t\big(\psi_t(x \mid s, a) \mid s, a\big), \ \psi_0(x \mid s, a) = x$$

$$\iff \psi_t(x \mid s, a) = x + \int_0^t v_\tau\big(\psi_\tau(x \mid s, a) \mid s, a\big) \, \mathrm{d}\tau.$$

We can solve this IVP using standard numerical integration techniques (Butcher, 2016). In doing so, we obtain an ODE-induced probability path defined as the pushforward $p_t := \psi_t(\cdot \mid S, A)_\sharp p_0(\cdot)$, i.e., the distribution of $\psi_t(X_0 \mid S, A)$ where $X_0 \sim p_0(\cdot)$. To ensure that $p_1$ coincides with the successor measure $m_\gamma^\pi$, Farebrother et al. (2025) propose to learn the parameterized vector field $v_t(\cdots ; \theta)$ by minimizing the TD-FLOW loss $\ell_{\text{TD-FLOW}}(\theta)$:

$$(1 - \gamma) \mathbb{E}_{\substack{S,A,S', \\ t, X_0, \vec{X}_t}} \left[ \left\| v_t(\vec{X}_t \mid S, A; \theta) - (S' - X_0) \right\|^2 \right] \quad (4)$$

$$+ \gamma \mathbb{E}_{\substack{S,A,S',A', \\ t, \overset{\frown}{X}_t}} \left[ \left\| v_t(\overset{\frown}{X}_t \mid S, A; \theta) - v_t(\overset{\frown}{X}_t \mid S', A'; \bar{\theta}) \right\|^2 \right],$$

where transitions $(S, A, S')$ are sampled from a dataset $\mathcal{D}$, $A' \sim \pi(\cdot \mid S')$, $t \sim \mathcal{U}([0,1])$, $X_0 \sim p_0$, $\vec{X}_t = (1 - t)X_0 + tS'$, $\overset{\frown}{X}_t = \psi_t(X_0 \mid S', A'; \bar{\theta})$, and $\bar{\theta}$ are non-trainable target parameters updated as a moving average of $\theta$. Recall that the Bellman equation (3) defines the successor measure as a mixture distribution with weights $1 - \gamma$ and $\gamma$. The TD-FLOW objective reflects this: the first term is a conditional flow-matching loss (Lipman et al., 2023) targeting the one-step transition kernel $P(\cdot \mid S, A)$, while the second is a marginal flow-matching term targeting the bootstrapped successor measure $m_\gamma^\pi(\cdot \mid S', A')$. Farebrother et al. (2025) show that jointly optimizing these components with the mixture weighting recovers the successor measure at convergence.

# 3. Planning via Geometric Policy Composition

In the sequel, we consider an agent equipped with a repertoire of pretrained policies $\{\pi_z\}_{z \in \mathsf{Z}}$ indexed by $z$ (e.g., a state for goal-conditioned policies or, more generally, a latent variable parameterizing diverse behaviors). Our objective is to learn a predictive model of the policies' behaviors that enables planning for arbitrary downstream tasks, without requiring further online interaction with the environment or additional fine-tuning.

To this end, we first formalize how GHM predictions compose to evaluate plans that stochastically switch among a subset of policies, showing how the successor measures of constituent policies combine to yield the successor measure of the composite policy. We then address the challenge of learning accurate GHMs across timescales by introducing a consistency objective that enforces coherence across

horizons, improving long-horizon predictions. Finally, with these tools we detail our compositional planning procedure.

## 3.1. Evaluating Geometric Switching Policies

A natural way to chain policies is through *geometric switching*: for each policy $\pi_{z_i}$ in a sequence, execution continues with probability $1 - \alpha_i \in [0, 1]$, or switches to policy $\pi_{z_{i+1}}$ with probability $\alpha_i$. Such a policy can be written as: $\nu := \pi_{z_1} \xrightarrow{\alpha_1} \pi_{z_2} \cdots \xrightarrow{\alpha_{n-1}} \pi_{z_n}$. By definition, the final policy $\pi_{z_n}$ is absorbing, meaning the agent commits to it for the remainder of the episode, so its switching probability is $\alpha_n = 0$. These non-Markovian policies are called Geometric Switching Policies (GSPs; Thakoor et al., 2022). The term "geometric" captures that each policy $\pi_{z_i}$ is followed for a geometrically distributed duration $T_i \sim \text{Geom}(\alpha_i)$.

To analyze these policies, we must understand how this switching mechanism interacts with the MDP's *global* discount factor, $\gamma$. Recall that $\gamma$ can be interpreted as the probability the episode continues to the next step, while $1 - \gamma$ gives the probability of halting. When following policy $\pi_{z_k}$ within a GSP, there are two reasons it might stop executing that policy: (1) the episode halts, with probability $1 - \gamma$; or (2) the policy switches to $\pi_{z_{k+1}}$, with probability $\alpha_k$. Thus, continuing to follow $\pi_{z_k}$ for one more step requires that neither event occur, which happens with probability $\beta_k := \gamma(1 - \alpha_k)$. This quantity acts as an effective discount factor for the duration spent executing $\pi_{z_k}$. Note that $\beta_n = \gamma$, since the final policy is absorbing.

We can now characterize the successor measure of a GSP. Note that the agent might reach a successor state $s^+$ through multiple paths: arriving while still executing $\pi_{z_1}$, or switching to $\pi_{z_2}$ and reaching $s^+$ from there, and so on. Each path contributes to the overall successor measure and must be weighted appropriately.

**Definition 1.** *Let* $\nu := \pi_{z_1} \xrightarrow{\alpha_1} \pi_{z_2} \cdots \xrightarrow{\alpha_{n-1}} \pi_{z_n}$ *be a geometric switching policy with global discount factor* $\gamma \in (0, 1)$ *and effective discount factors* $\beta_k := \gamma(1 - \alpha_k)$ *for* $k \in [\![n]\!]$. *The weight of the $k$-th policy is*

$$w_k := \frac{1 - \gamma}{1 - \beta_k} \prod_{i=1}^{k-1} \frac{\gamma - \beta_i}{1 - \beta_i},$$

*where an empty product equals* $1$ *(hence* $w_1 = \frac{1-\gamma}{1-\beta_1}$*).*

These weights capture the relative contribution of each policy phase to the successor measure of the GSP. Intuitively, $w_k$ reflects the probability that the agent (i) survives the first $k - 1$ policy phases without the episode halting, and (ii) reaches states under policy $\pi_{z_k}$ rather than having already switched to a later policy. With these weights, we define the successor measure of a GSP in the following result.

**Theorem 1.** *Let* $\nu := \pi_{z_1} \xrightarrow{\alpha_1} \pi_{z_2} \cdots \xrightarrow{\alpha_{n-1}} \pi_{z_n}$ *be a geometric switching policy with global discount factor* $\gamma \in (0,1)$, *effective discount factors* $\{\beta_k\}_{k=1}^n$, *and weights* $\{w_k\}_{k=1}^n$ *from Definition 1. For any state-action pair* $(s,a)$, *the successor measure of* $\nu$ *decomposes as:*

$$
m_\gamma^\nu(\mathrm{d}s' \mid s,a) = \sum_{k=1}^n w_k \int_{\substack{s_1,\ldots,s_{k-1} \\ a_1,\ldots,a_{k-1}}} m_{\beta_1}^{\pi_{z_1}}(\mathrm{d}s_1 \mid s,a)
$$
$$
\times \pi_{z_2}(\mathrm{d}a_1 \mid s_1) \cdots m_{\beta_k}^{\pi_{z_k}}(\mathrm{d}s' \mid s_{k-1}, a_{k-1}).
$$

This theorem decomposes the successor measure of a GSP as a mixture distribution with $n$ components. The $k$-th component, weighted by $w_k$, captures the state distribution under policy $\pi_{z_k}$, having passed through intermediate states visited under $\pi_{z_1}, \ldots, \pi_{z_{k-1}}$[3]. Such a decomposition yields a strategy for estimating action-value function similar to (2): sample a state from each component, evaluate its reward, and form a weighted sum using $w_k$. Rather than sampling from each component independently, we leverage their overlapping sequential structure to draw samples more efficiently. We achieve this through composition: starting from $(s,a)$, we sample $S_1^+ \sim m_{\beta_1}^{\pi_{z_1}}(\cdot \mid s,a)$, then use $S_1^+$ to sample $S_2^+ \sim m_{\beta_2}^{\pi_{z_2}}(\cdot \mid S_1^+, A_1^+)$ where $A_1^+ \sim \pi_{z_2}(\cdot \mid S_1^+)$, and so on. As the following lemma formalizes, the weighted sum $\sum_k w_k \, r(S_k^+)$ yields an unbiased estimator of $Q_\gamma^\nu$.

**Lemma 1.** *Let* $\nu := \pi_{z_1} \xrightarrow{\alpha_1} \pi_{z_2} \cdots \xrightarrow{\alpha_{n-1}} \pi_{z_n}$ *be a geometric switching policy with global discount factor* $\gamma \in (0,1)$, *effective discount factors* $\{\beta_k\}_{k=1}^n$, *and weights* $\{w_k\}_{k=1}^n$ *from Definition 1. For any reward function* $r : \mathsf{S} \to \mathbb{R}$ *and state-action pair* $(s,a)$, *set* $(S_0^+, A_0^+) = (s,a)$ *and for* $k = 1, \ldots, n$ *sample* $S_k^+ \sim m_{\beta_k}^{\pi_{z_k}}(\cdot \mid S_{k-1}^+, A_{k-1}^+)$ *and* $A_k^+ \sim \pi_{z_{k+1}}(\cdot \mid S_k^+)$. *Then the single-sample monte-carlo estimator*

$$
\widehat{Q}_\gamma^\nu := (1-\gamma)^{-1} \sum_{k=1}^n w_k \, r(S_k^+),
$$

*is an unbiased estimate of* $Q_\gamma^\nu$, *i.e.,* $\mathbb{E}\big[\widehat{Q}_\gamma^\nu\big] = Q_\gamma^\nu(s,a)$ *where the expectation is over the joint distribution of* $(S_1^+, A_1^+, \ldots, S_n^+)$ *induced by the sampling procedure.*

This lemma generalizes several previous results: it extends Thakoor et al. (2022, Theorem 3.2), which assumed fixed switching probabilities, i.e., $\alpha_k = \alpha, \forall k \in [\![n-1]\!]$, and further generalizes results in Janner et al. (2020), that additionally impose a fixed policy throughout, i.e., $\pi_{z_k} = \pi, \forall k \in [\![n]\!]$. By allowing both policies and switching probabilities to vary, this result enables the evaluation of a broader class of policies and brings us closer to the options framework (Sutton, 1995; Sutton et al., 1999; Precup, 2000).

---

[3]Each component is an application of the Chapman–Kolmogorov equation (Chapman, 1928; Kolmogoroff, 1931), retaining kernel composition via marginalization but replacing one-step dynamics $P(\cdot \mid s,a)$ with jumpy dynamics via $m_\beta^\pi(\cdot \mid s,a)$.

## 3.2. Learning Geometric Horizon Models Across Multiple Timescales

A core requirement of our planning framework is the ability to predict the behavior of many policies over multiple timescales, enabling us to calculate the expected return of candidate geometric switching policies. A natural extension of the TD-FLOW objective (4) conditions the vector field $v$ on both the policy encoding $z$ and discount factor $\gamma$, yielding a single unified model across policies and horizons. However, generalizing across many horizons is challenging: variance increases with horizon length, reducing per-horizon accuracy and destabilizing training (Petrik & Scherrer, 2008).

To address this challenge, we propose a generalization of TD-FLOW that enforces consistency across horizons. Rather than learning each horizon independently, we exploit a Bellman-like relationship between the successor measure at two discount factors $\beta \leq \gamma$ to bootstrap longer-horizon predictions from shorter-horizon ones:

$$
m_\gamma^\pi(\cdot \mid s,a) = w_1 \, P(\cdot \mid s,a) \tag{5}
$$
$$
+ w_2 \, \mathbb{E}_{S' \sim P(\cdot \mid s,a), A' \sim \pi(\cdot \mid S')} \big[ m_\beta^\pi(\cdot \mid S', A') \big]
$$
$$
+ w_3 \, \mathbb{E}_{\substack{S' \sim P(\cdot \mid s,a), A' \sim \pi(\cdot \mid S') \\ S^+ \sim m_\beta^\pi(\cdot \mid S', A'), A^+ \sim \pi(\cdot \mid S^+)}} \big[ m_\gamma^\pi(\cdot \mid S^+, A^+) \big],
$$

with weights $w_1 = (1-\gamma)$, $w_2 = \gamma\frac{1-\gamma}{1-\beta}$, and $w_3 = \gamma\frac{\gamma-\beta}{1-\beta}$. This follows directly from Theorem 1 by considering the switching policy $\nu := \pi \xrightarrow{\alpha_1 = 1} \pi \xrightarrow{\alpha_2 = 1 - \beta/\gamma} \pi$ and noting that $m_{\gamma(1-\alpha_1)}^\pi = m_0^\pi = P$. Building on this result, we extend the derivation of TD-FLOW from Farebrother et al. (2025) to the new Bellman equation in (5) (full derivation in Appendix C.1), arriving at what we call the *Temporal Difference Horizon Consistency (*TD-HC*) loss* $\ell_{\text{TD-HC}}(\theta; \beta, \gamma)$:

$$
w_1 \, \mathbb{E}_{\substack{S,A,S' \\ t, X_0, \vec{X}_t}} \big\| v_t(\vec{X}_t \mid S, A, \gamma; \theta) - (S' - X_0) \big\|^2 + \tag{6}
$$
$$
w_2 \, \mathbb{E}_{\substack{S,A,S',A' \\ t, \widetilde{X}_t^\beta}} \big\| v_t(\widetilde{X}_t^\beta \mid S, A, \gamma; \theta) - v_t(\widetilde{X}_t^\beta \mid S', A', \beta; \bar\theta) \big\|^2 +
$$
$$
w_3 \, \mathbb{E}_{\substack{S,A,S^+,A^+ \\ t, \widetilde{X}_t^\gamma}} \big\| v_t(\widetilde{X}_t^\gamma \mid S, A, \gamma; \theta) - v_t(\widetilde{X}_t^\gamma \mid S^+, A^+, \gamma; \bar\theta) \big\|^2,
$$

where $\widetilde{X}_t^\beta \sim \psi_t(\cdot \mid S', A', \beta; \bar\theta)_{\#} p_0(\cdot)$, $S^+ \sim m_\beta^\pi(\cdot \mid S', A')$, $A^+ \sim \pi(\cdot \mid S^+)$ and $\widetilde{X}_t^\gamma \sim \psi_t(\cdot \mid S^+, A^+, \gamma; \bar\theta)_{\#} p_0(\cdot)$. When $\gamma = \beta$, the third term vanishes, and we recover the original TD-FLOW loss. In practice, we sample effective horizons $\tau_\gamma \sim \mathcal{U}[1/1-\gamma_{\min}, 1/1-\gamma_{\max}]$ and $\tau_\beta \sim \mathcal{U}[1/1-\gamma_{\min}, \tau_\gamma]$, and convert them to discount factors via $\gamma = 1 - 1/\tau_\gamma$ and $\beta = 1 - 1/\tau_\beta$. Additionally, we only apply horizon consistency (i.e., $\beta \neq \gamma$) to a small proportion of each mini-batch. This is motivated by the fact that the consistency term requires sampling from the model's own predictions at horizon $\beta$ and using these samples as conditioning for the longer horizon $\gamma$, meaning errors in the

model's current predictions can compound. Restricting the consistency term to a small fraction of the batch, we gain the benefits of horizon alignment while ensuring the majority of updates come from TD-FLOW. This design follows standard practice for consistency-like generative modeling (e.g., Frans et al., 2025; Geng et al., 2025; Boffi et al., 2025).

### 3.3. Compositional Planning

With the machinery for training policy-conditioned GHMs across multiple timescales now in place, we can turn to the central question: how to use them to solve downstream tasks? Given a reward function $r : \mathsf{S} \to \mathbb{R}$, our goal is to find a sequence of policies that maximizes expected return. Since the learned GHMs already capture the full complexity of how policies evolve in the environment, evaluating $Q_\gamma^\nu$ from Lemma 1 requires only specifying the policy embeddings $z_1, \ldots, z_n$, hence planning reduces to the following optimization problem:

$$\max_{a_1, z_1, \ldots, z_n} Q_\gamma^{\pi_{z_1} \xrightarrow{\alpha_1} \pi_{z_2} \cdots \xrightarrow{\alpha_{n-1}} \pi_{z_n}} (s, a_1). \quad (7)$$

The switching probabilities $\{\alpha_i\}$ control how long each policy executes before transitioning to the next, and are treated as hyperparameters. Once the optimal sequence $(a_1^*, z_1^*, \ldots, z_n^*)$ is identified, we execute the action $a_1^*$ followed by policy $\pi_{z_1^*}$, replanning at future states as needed.

Notably, this planning objective unifies several existing approaches as special cases. By varying the switching probabilities $\{\alpha_i\}$, one can interpolate between action-level control and planning over policies by:

- Setting $\alpha_1 = \cdots = \alpha_n = 1$, which reduces to optimizing over sequences of primitive actions, equivalent to Model-Predictive Control with horizon $n$.

- Setting $\alpha_1 = 1$ and $\alpha_2 = \cdots = \alpha_n = 0$, yielding *Generalized Policy Improvement* (*GPI*; Barreto et al., 2017).

- Setting $\alpha_1 = \cdots = \alpha_{n-1} = \alpha$ for some fixed $\alpha \in (0, 1)$, which recovers *Geometric Generalized Policy Improvement* (*GGPI*; Thakoor et al., 2022).

We refer to our approach – which allows distinct switching probabilities $\alpha_1, \ldots, \alpha_{n-1} \in (0, 1)$ – as COMPPLAN. Likewise, we refer to action-level planning as ACTION-PLAN. Crucially, the same pretrained GHMs power all of these methods. By conditioning on policies and a continuum of timescales, our framework spans one-step world models ($\alpha = 1$) through long-horizon policy composition ($\alpha_i \in (0, 1)$), unifying previously disparate paradigms.

**Optimization via random shooting.** The maximization in (7) is tractable when policies are indexed by a finite set, but becomes challenging for large or continuous $\mathsf{Z}$. In this paper, we focus on goal-conditioned policies where $\mathsf{Z} = \mathsf{S}$ and $z \in \mathsf{S}$ represents a subgoal. Here, the key difficulty lies in proposing good candidate subgoals without searching over the entire state space. Our solution is to use the GHMs themselves as a proposal distribution. Given a goal $g \in \mathsf{S}$, we generate waypoints from $z_0 := s$ by composing $m_{\beta_i}^{\pi_g}$ over horizons $\{\beta_i\}$ as:

$$a_i \sim \pi_g(\cdot \mid z_i), \quad z_{i+1} \sim m_{\beta_{i+1}}^{\pi_g}(\cdot \mid z_i, a_i),$$
$$\text{for } i = 0, \ldots, n-1.$$

This produces a sequence of subgoals $(z_1, \ldots, z_n)$ that naturally guides progress towards the final objective. Alternatively, we can sample from an unconditional GHM that predicts plausible successor states under the data distribution (i.e., the behavior policy). To enable this, we stochastically mask the policy encoding ($z = \varnothing$) during training, bootstrapping with the dataset action in (4) when masked.

With a proposal distribution, planning reduces to random shooting (Matyáš, 1965): we (1) sample $m$ candidate sequences $\{(z_1^{(i)}, \ldots, z_n^{(i)})\}_{i=1}^m$ from the proposal distribution; (2) evaluate $Q_\gamma^{\nu^{(i)}}(s, a_1^{(i)})$ for each candidate switching policy $\nu^{(i)} := \pi_{z_1^{(i)}} \xrightarrow{\alpha_1} \pi_{z_2^{(i)}} \cdots \xrightarrow{\alpha_{n-1}} \pi_{z_n^{(i)}}$ using Lemma 1 where $a_1^{(i)} \sim \pi_{z_1^{(i)}}(\cdot \mid s)$; and (3) select the sequence $(a_1^*, z_1^*, \ldots, z_n^*)$ with the highest value $Q_\gamma^{\nu^*}(s, a_1^*)$. The full method is summarized in Algorithm 2.

## 4. Experiments

Our empirical evaluation tests the core hypothesis of this work: that learning a jumpy world model over a diverse collection of parameterized policies enables effective and efficient compositional planning. We begin by describing the experimental setting and the training procedure for the base policies. We subsequently compare the performance of these policies against that of compositional planning over them. Additionally, we benchmark COMPPLAN against other test-time planning approaches and hierarchical methods. Finally, we examine the effect of the proposed consistency loss on both model accuracy and planning performance. Additional ablations on the replanning frequency, planning objective, and proposal distribution are provided in Appendix D.

### 4.1. Experimental Setup

**Benchmark and Dataset** All experiments use the OG-Bench benchmark (Park et al., 2025a), which provides challenging long-horizon robotic manipulation and locomotion tasks structured as offline goal-conditioned reinforcement learning problems. We focus on ant navigation tasks across different maze topologies (MEDIUM, LARGE, and GIANT) as well as multi-cube robotic manipulation. For both policy and GHM training, we use the *standard* NAVIGATE and PLAY datasets for ANTMAZE and CUBE, respectively.

**Base Policies** The effectiveness of compositional planning depends on the quality and characteristics of the chosen policies. To explore this, we train five policy types, each exhibiting different tradeoffs between GHM learning and planning performance: 1) Goal-Conditioned TD3 (GC-TD3; Pirotta et al., 2024); 2) Goal-Conditioned 1-Step RL (GC-1S); 3) Contrastive RL (CRL; Eysenbach et al., 2022); 4) Goal-Conditioned Behavior Cloning (GC-BC; Lynch et al., 2019; Ghosh et al., 2021); and 5) Hierarchical Flow Behavior Cloning (HFBC; Park et al., 2025b). Additional implementation details are provided in Appendix F.1.

### 4.2. How Do Policies Affect Compositional Planning?

For each policy family, we train a GHM in an off-policy manner using the TD-HC loss described in §3.2. GHMs are trained for 3M gradient steps using the Adam optimizer (Kingma & Ba, 2015) with a batch size of 256. The model architecture follows a U-Net-style design, similar to Farebrother et al. (2025). Both the timestep $t$ and the discount factor $\gamma$ are embedded by first applying a sinusoidal embedding to increase dimensionality, followed by a two-layer MLP with mish activations (Misra, 2019). For the discount embedding, we further concatenate the vector $[\gamma, 1 - \gamma, -\log(1 - \gamma)]$, where $-\log(1 - \gamma)$ corresponds to the logarithm of the effective horizon; we find this improves the model's sensitivity to the discount factor. Other conditioning information, such as the state-action pair and policy embedding $z$, is processed through an additional MLP and added to both the time and discount embeddings. The network incorporates conditioning information via FiLM modulation (Perez et al., 2018). When training each GHM, we apply the horizon-consistency objective (6) (i.e., $\beta \neq \gamma$) to 25% of each mini-batch in ANTMAZE and 12.5% in CUBE. We also train the unconditional model (i.e., $z = \varnothing$) for 10% of each mini-batch; these two proportions do not overlap.

During evaluation, we tailor the proposal distribution to each domain based on their characteristics. In ANTMAZE, states are separated by large temporal distances and physical barriers, so an unconditional proposal would waste most samples on irrelevant regions of the state space; we therefore sample 256 subgoal sequences from the goal-conditioned GHM. In CUBE, tasks consist of short pick-and-place sequences with many viable paths, making the unconditional GHM a natural fit; we sample 1024 sequences to cover the broader proposal. See Appendix D.5 for ablations on these choices.

We begin by comparing the zero-shot performance of the base policies against our compositional planning approach. Table 1 details these results (averaged over three seeds), showing that compositional planning consistently improves upon the zero-shot policies. This demonstrates the effectiveness of our method in selecting policy sequences that exceed the performance of the best individual policy. While improvements are evident across all domains, the gains are particularly pronounced in complex, long-horizon tasks such as ANTMAZE-GIANT and CUBE-{3,4}, where success rates can rise from 10% to 90% in the most extreme cases.

Among the evaluated policy classes, HFBC emerges as the most consistent zero-shot performer. COMPPLAN further improves HFBC, notably in ANTMAZE-GIANT and CUBE-4. CRL, in contrast, is effective in ANTMAZE but underperforms in CUBE. We hypothesize this stems from the inductive bias in CRL's representation learning, which approximates the goal-conditioned value function as $Q(s, a, g) \approx \phi(s, a)^\top \psi(g)$. This factorization effectively captures the ant's spatial position – yielding robust navigation in ANTMAZE – but fails to encode complex object-related features, resulting in weaker CUBE policies. Notably, incorporating planning not only improves upon CRL's strong baseline in ANTMAZE, but also achieves non-trivial success rates in CUBE, demonstrating that our approach can extract utility from base policies otherwise limited by their inductive bias.

Interestingly, the other policies exhibit the opposite trend. When evaluated zero-shot, they show weak performance in medium to long-horizon tasks – ANTMAZE-MEDIUM, ANTMAZE-LARGE, and CUBE tasks all prove challenging, with CUBE success rates falling below $10 - 15\%$. However, these policies prove remarkably effective when integrated into our compositional planning framework, with success rates climbing above 70% in many tasks. Taken together, these results suggest that zero-shot metrics tell us little about how well a policy will compose; its utility may only become clear when orchestrated with other policies.

### 4.3. How Does Compositional Planning Compare to Other Planning and Hierarchical Approaches?

Our results in §4.2 demonstrate that compositional planning unlocks capabilities beyond what any single policy achieves alone. We now situate our approach among existing planning and hierarchical methods, revealing that COMPPLAN's advantages stem from its unique combination of temporal abstraction and flexible composition.

Recall from §3.3 that different choices of switching probabilities recover existing methods as special cases. This observation suggests a natural ablation: comparing COMPPLAN against these special cases can disentangle the contribution of its key components. At one extreme lies *Generalized Policy Improvement* (GPI; Barreto et al., 2017) which sets $\alpha_1 = 1$ and commits to a single policy for the remainder of the episode. GPI leverages our GHMs to estimate each policy's value, selecting the best in-class policy at each timestep according to:

$$\max_{z, A \sim \pi_z(\cdot|s)} Q_\gamma^{\pi_z}(s, a) = (1 - \gamma)^{-1} \mathbb{E}_{S \sim m_\gamma^{\pi_z}(\cdot|s,a)} \left[ r(S) \right].$$

*Table 1.* Success rate (↑) of base policies $\pi_g$ (Zero Shot) and compositional planning with GHMs (COMPPLAN; ours) averaged over tasks. We report the mean and standard deviation over 3 seeds. We highlight relative increases and decreases in performance w.r.t. the base policies. Additionally, we **bold** the best performance for each domain.

| Domain | CRL | | GC-1S | | GC-BC | | GC-TD3 | | HFBC | |
|---|---|---|---|---|---|---|---|---|---|---|
| | Zero Shot | COMPPLAN | Zero Shot | COMPPLAN | Zero Shot | COMPPLAN | Zero Shot | COMPPLAN | Zero Shot | COMPPLAN |
| ANTMAZE-MEDIUM | 0.88 | **0.97 (0.02)** | 0.56 | 0.87 (0.05) | 0.49 | 0.85 (0.08) | 0.65 | 0.65 (0.03) | 0.94 | 0.94 (0.01) |
| ANTMAZE-LARGE | 0.84 | 0.90 (0.00) | 0.21 | 0.61 (0.04) | 0.18 | 0.73 (0.02) | 0.23 | 0.48 (0.05) | 0.78 | **0.92 (0.02)** |
| ANTMAZE-GIANT | 0.16 | 0.29 (0.03) | 0.00 | 0.02 (0.00) | 0.00 | 0.03 (0.01) | 0.00 | 0.01 (0.01) | 0.42 | **0.79 (0.04)** |
| CUBE-1 | 0.28 | 0.86 (0.02) | 0.37 | 0.66 (0.02) | 0.90 | **0.99 (0.01)** | 0.58 | 0.91 (0.01) | 0.80 | 0.97 (0.01) |
| CUBE-2 | 0.02 | 0.50 (0.03) | 0.10 | 0.57 (0.09) | 0.15 | **0.97 (0.01)** | 0.12 | 0.82 (0.01) | 0.76 | 0.77 (0.02) |
| CUBE-3 | 0.01 | 0.73 (0.02) | 0.01 | 0.67 (0.02) | 0.09 | **0.92 (0.01)** | 0.12 | 0.83 (0.04) | 0.64 | 0.83 (0.03) |
| CUBE-4 | 0.00 | 0.39 (0.04) | 0.01 | 0.60 (0.02) | 0.00 | **0.76 (0.03)** | 0.00 | 0.57 (0.03) | 0.24 | 0.67 (0.03) |

GPI thus serves as a test of whether composing policies offers an advantage over merely selecting among them.

At the other extreme, we compare against action-level planning (ACTIONPLAN), which sets $\alpha_1 = \cdots = \alpha_n = 1$ and operates at the granularity of individual actions rather than policies. To fairly isolate the effect of temporal abstraction, we train a dedicated one-step world model $\tilde{p}(\cdot|s,a)$ using the same flow-matching framework, network architecture, and training procedure as our GHMs – differing only in the removal of policy and discount conditioning. Given this model, we optimize the following objective:

$$\arg\max_{A_1,\ldots,A_n} \sum_{k=1}^{n} \gamma^k r(S_{k+1}),$$

where $S_{k+1} \sim \tilde{p}(\cdot \mid S_k, A_k)$ and $A_k \sim \pi_g(\cdot \mid S_k)$. This comparison directly tests whether planning over sequences of policies – enabled by jumpy predictions – confers an advantage over action-level planning.

Figure 1 reveals a clear pattern: on long-horizon tasks, COMPPLAN substantially outperforms both alternatives, achieving an $89\%$ relative improvement over GPI and a $201\%$ gain over ACTIONPLAN (averaged across policies and long-horizon domains). These gains indicate that neither policy selection alone nor action-level planning captures the full benefit of our compositional framework. Rather, the combination of planning over sequences of policies at multiple timescales unlocks strong long-horizon capabilities.

Furthermore, we compare against methods that learn hierarchical structure during training: HIQL (Park et al., 2023) and SHARSA (Park et al., 2025b), the current state-of-the-art on OGBench. These approaches train high-level policies to select subgoals or skills, whereas COMPPLAN performs composition at test time using pre-trained GHMs. Table 2 presents this comparison where COMPPLAN employs HFBC base policies similar to SHARSA. Our approach consistently outperforms both hierarchical methods, with the largest margins on the most challenging tasks. In CUBE-4, COMPPLAN achieves $67\%$ success compared to $9\%$ for SHARSA and $0\%$ for HIQL – demonstrating that

*Table 2.* Success rate (↑) of hierarchical baselines and compositional planning with HFBC policies (COMPPLAN; ours). For each domain, we highlight the best performance.

| Domain | HIQL | SHARSA | HFBC | COMPPLAN |
|---|---|---|---|---|
| ANTMAZE-MEDIUM | 0.96 (0.01) | 0.91 (0.03) | 0.94 | 0.94 (0.01) |
| ANTMAZE-LARGE | 0.91 (0.02) | 0.88 (0.03) | 0.78 | 0.92 (0.02) |
| ANTMAZE-GIANT | 0.65 (0.05) | 0.56 (0.07) | 0.42 | 0.79 (0.04) |
| CUBE-1 | 0.15 (0.03) | 0.70 (0.03) | 0.84 | 0.97 (0.01) |
| CUBE-2 | 0.06 (0.02) | 0.60 (0.07) | 0.70 | 0.77 (0.02) |
| CUBE-3 | 0.03 (0.01) | 0.50 (0.09) | 0.54 | 0.83 (0.03) |
| CUBE-4 | 0.00 (0.00) | 0.09 (0.04) | 0.34 | 0.67 (0.03) |

*Table 3.* Accuracy (EMD; ↓) of GHMs trained with our horizon consistency loss (TD-HC) and without (TD-FLOW) for discount factor $\gamma = 0.995$. We highlight the best performing method.

| Domain | CRL | | GC-1S | |
|---|---|---|---|---|
| | TD-FLOW (✗) | TD-HC (✓) | TD-FLOW (✗) | TD-HC (✓) |
| ANTMAZE-MEDIUM | 4.41 (0.05) | 4.22 (0.06) | 4.40 (0.02) | 4.22 (0.03) |
| ANTMAZE-LARGE | 5.24 (0.07) | 4.81 (0.03) | 5.12 (0.18) | 4.67 (0.04) |
| ANTMAZE-GIANT | 6.77 (0.49) | 5.74 (0.06) | 7.29 (0.69) | 5.25 (0.08) |
| CUBE-1 | 1.60 (0.02) | 1.57 (0.03) | 1.43 (0.00) | 1.33 (0.03) |
| CUBE-2 | 2.36 (0.03) | 2.23 (0.02) | 1.86 (0.04) | 1.71 (0.01) |
| CUBE-3 | 2.15 (0.02) | 2.10 (0.02) | 1.80 (0.04) | 1.71 (0.03) |
| CUBE-4 | 2.41 (0.03) | 2.34 (0.01) | 2.13 (0.03) | 2.05 (0.03) |

test-time composition can surpass learned hierarchies when tasks demand flexible, long-horizon reasoning. Notably, COMPPLAN requires no task-specific training, suggesting a promising alternative to task-specific hierarchical methods.

### 4.4. How Does Horizon Consistency Affect GHM Learning?

Having established that compositional planning yields strong empirical gains, we now turn inward to examine one of our methodological contributions: the horizon consistency objective from §3.2. We investigate its impact through two complementary lenses – generative fidelity and downstream planning – revealing that consistency plays distinct roles at different stages of the pipeline.

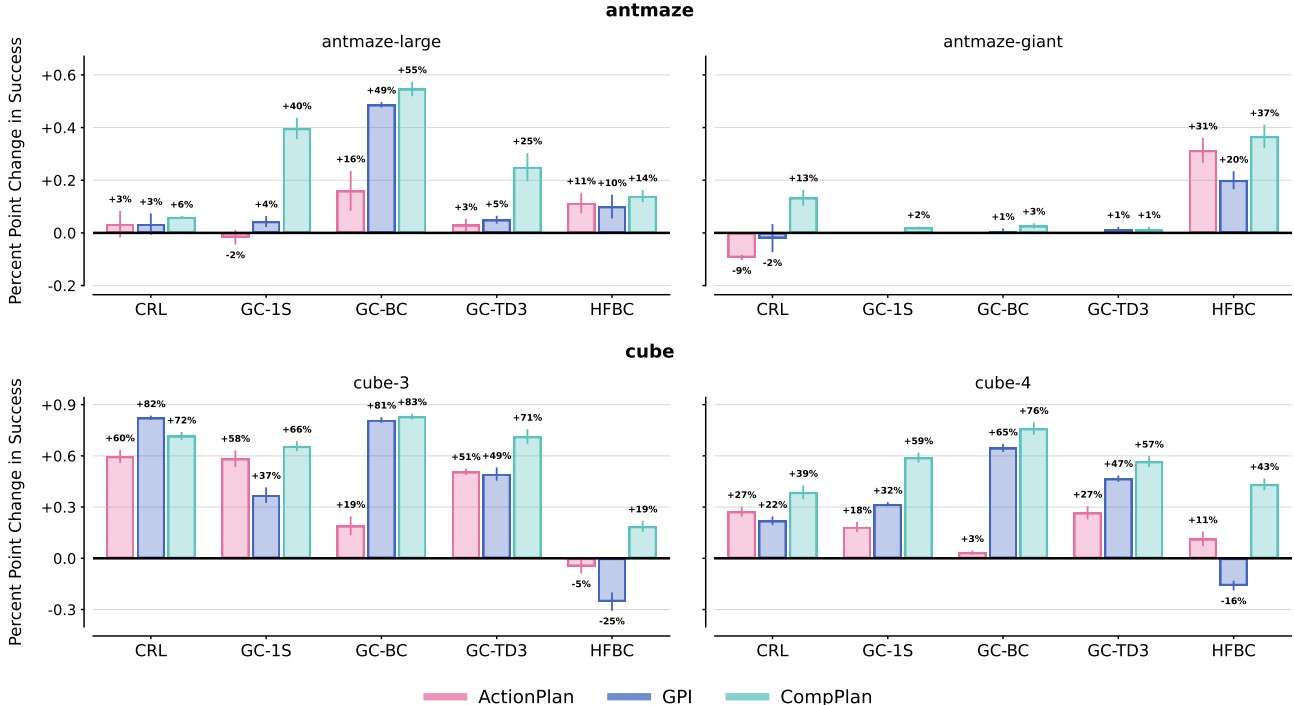

*Figure 1.* Percentage point change ($\uparrow$) over zero-shot policies. We compare: action-level planning (ACTIONPLAN) with a world model; generalized policy improvement (GPI) and compositional planning (COMPPLAN; ours) with GHMs.

We begin by asking whether enforcing consistency across timescales produces more accurate GHM predictions. To isolate this effect, we train two GHMs – one using TD-FLOW and the other using TD-HC – keeping all else fixed. We obtain ground-truth samples by executing 64 policy rollouts from 256 randomly selected (state, goal) pairs and resampling 2048 visited states according to $t \sim \text{Geom}(1 - \gamma)$. We then draw an equal number of samples from each GHM and compute the Earth Mover's Distance (EMD; Rubner et al., 2000) between the two sets. As Table 3 demonstrates, consistency systematically improves accuracy at long horizons. The effect is particularly striking in ANTMAZE, where the maze structure imposes hard constraints on reachability. For example, employing TD-HC with GC-1S policies in ANTMAZE-GIANT results in a $28\%$ reduction in EMD. Qualitatively, Figure 10 shows that TD-HC on ANTMAZE-GIANT leads to fewer samples erroneously traversing walls, a failure mode that compounds over long horizons.

Given these accuracy improvements, one might expect correspondingly large gains in planning. Surprisingly, Table 9 tells a different story: planning success rates are nearly identical with and without consistency, averaging only a $5\%$ relative improvement. These findings are not contradictory but rather illuminate when consistency matters most. Our planning procedure evaluates candidate sequences using effective horizons $\{\beta_i\}$ in the range of $50 - 100$ steps (i.e., $\beta_i \in [0.98, 0.99]$), not the $200+$ step horizons where

consistency provides its largest accuracy gains. At these moderate timescales, the base TD-FLOW objective already learns sufficiently accurate models to rank policies correctly. The consistency objective thus provides a margin of safety for long-horizon predictions without being strictly necessary for the planning horizons we employ in OGBench. This suggests that practitioners facing longer-horizon tasks would benefit most from the consistency loss herein.

## 5. Discussion

This work reframes pre-trained policies not as isolated controllers but as composable primitives – building blocks to be sequenced. Jumpy world models provide the mechanism: by predicting successor states for many policies across a continuum of horizons, they enable planning over behavior rather than primitive actions. Empirically, compositional planning consistently outperforms individual policies, hierarchical methods, and action-level planning, with striking gains at long horizons. Looking ahead, we see many promising directions: learning state-dependent switching probabilities, jointly learning policies and predictive models, employing more sample-efficient model-predictive control methods, and exploring jumpy world models in learned latent spaces.

## Impact Statement

This paper presents work whose goal is to advance the field of machine learning. There are many potential societal consequences of our work, none of which we feel must be specifically highlighted here.

## Acknowledgements

The authors thank Harley Wiltzer, Arnav Jain, Pierluca D'oro, Nate Rahn, Michael Rabbat, Yann Ollivier, Marlos C. Machado, Michael Bowling, Adam White, and Doina Precup for useful discussions that helped improve this work. MGB is supported by the Canada CIFAR AI Chair program and NSERC. Finally, this work was made possible by open-source software, particularly MuJoCo (Todorov et al., 2012) and Python libraries such as NumPy (Harris et al., 2020), Matplotlib (Hunter, 2007), Seaborn (Waskom, 2021), and Einops (Rogozhnikov, 2022).

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

# Appendices

## A. Extended Related Works

**Successor Measure** Methods that learn (discounted) state occupancies employing temporal difference learning (Sutton, 1995) date back to the successor representation (Dayan, 1993) with more recent extensions like successor features (Barreto et al., 2017) and the successor measure (Blier et al., 2021; Blier, 2022). Janner et al. (2020) was the first to introduce a generative model of the successor measure with $\gamma$-models also referred to as geometric horizon models (Thakoor et al., 2022). Wiltzer et al. (2024b) additionally introduced $\delta$-models that learn a distribution over $\gamma$-models enabling applications in distributional RL (Bellemare et al., 2017; 2023; Dabney et al., 2018). Many generative modeling techniques have been applied to learn these models, including GANs (Janner et al., 2020; Wiltzer et al., 2024b), normalizing flows (Janner et al., 2020), VAEs (Thakoor et al., 2022; Tomar et al., 2024), flow matching (Farebrother et al., 2025; Zheng et al., 2026), and diffusion (Schramm & Boularias, 2024; Farebrother et al., 2025). Closely related is work on distributional successor features also known as multi-variate distributional RL (Freirich et al., 2019; Gimelfarb et al., 2021; Zhang et al., 2021; Wu et al., 2023; Wiltzer et al., 2024a; Zhu et al., 2024), that involves modeling the distribution over cumulative finite-dimensional features induced by a policy. From a generative modeling perspective, our work generalizes temporal difference flows (Farebrother et al., 2025) and shows how long-horizon predictions can be improved by training across multiple timescales with a novel horizon consistency objective.

Additionally, our compositional framework can be viewed as a generalization of Geometric Generalized Policy Improvement (GGPI; Thakoor et al., 2022) to arbitrary switching probabilities; however, our concrete formulation and empirical evaluation differ substantially from Thakoor et al. (2022). First, Thakoor et al. (2022) considers only four pre-trained policies in practice, and learn two separate GHMs with effective horizons of 5 and 10 steps respectively. Their experiments also restrict composition to sequences of only two policies. In contrast, we learn GHM models conditioned on a continuous family of policies and timescales with horizons up to $25\times$ longer, and evaluate GSPs with lengths ranging from 3 to 24 policies. As a result, our work not only enables a richer class of geometric switching policies but also performs a more comprehensive empirical validation of these techniques on challenging long-horizon tasks where temporal abstraction is most important.

**Planning With Temporal Abstractions** Several prior works have explored planning over subgoals or waypoints to solve long-horizon tasks (e.g., Nasiriany et al., 2019; Eysenbach et al., 2019; Nair & Finn, 2020; Chane-Sane et al., 2021; Fang et al., 2022; Hafner et al., 2022; Lo et al., 2024; Gürtler & Martius, 2025). These methods typically involve learning sub-goal conditioned policies together with a high-level dynamics model that predicts the outcomes of reaching these subgoals $k$ steps in the future, then employ model predictive control to select subgoal sequences. In contrast, GHMs model the entire state-occupancy distribution rather than a fixed $k$-step lookahead, allowing planning over arbitrary reward functions rather than just goal-conditioned tasks.

A parallel line of work employs diffusion models (Vincent, 2011; Sohl-Dickstein et al., 2015; Ho et al., 2020; Song et al., 2021; Lai et al., 2025) for trajectory-level planning (e.g., Janner et al., 2022; Ajay et al., 2023; Zheng et al., 2023; Chen et al., 2025; Yoon et al., 2025a;b; Luo et al., 2025; Lee & Choi, 2025). Rather than modeling policy-induced dynamics, these methods train generative models over trajectory segments and perform planning within the denoising process. Execution then relies on inverse dynamics models to extract actions from planned state sequences. While powerful, this paradigm has notable limitations: (i) it requires learning accurate inverse dynamics; (ii) planning quality depends heavily on the trajectory distribution in the training data rather than on the capabilities of any particular policy; and (iii) these methods often assume access to oracle goal representations during the denoising process. In contrast, our work is policy-grounded: it directly composes the behaviors of pre-trained policies by predicting their induced state occupancies, rather than planning over abstract trajectory segments. This makes our approach agnostic to the policy class and avoids inverse dynamics entirely since actions are sampled directly from the base policies.

Closer to our approach, are methods that learn dynamics models over temporally extended behaviors (e.g., Xie et al., 2021; Shi et al., 2022; Zhang et al., 2023; Mishra et al., 2023; Gürtler & Martius, 2025). These methods learn general latent skills together with a high-level dynamics model predicting the outcomes of their execution (more precisely the states reached a fixed number of steps in the future), and use MPC to plan over these skills. However, these approaches suffer from the same limitations as the $k$-step subgoal methods discussed above. While we only report experiments for goal-based policies, COMPPLAN also enables planning on top of any set of parameterized skills or policies, for example, those learned via unsupervised RL methods (Borsa et al., 2019; Touati & Ollivier, 2021; Touati et al., 2023; Park et al., 2024; Frans et al.,

2024; Cetin et al., 2025; Agarwal et al., 2025; Tirinzoni et al., 2025; Sikchi et al., 2025b;a; Bagatella et al., 2026).

Finally, our work connects to both the options framework (Sutton et al., 1999; Precup, 2000) and hierarchical RL (Schmidhuber, 1991; Kaelbling, 1993a; Parr & Russell, 1997; Barto & Mahadevan, 2003; Klissarov et al., 2025), which share a focus on temporal abstraction and multi-level decision making. Prior work has explored planning with options (e.g., Silver & Ciosek, 2012; Jinnai et al., 2019; Barreto et al., 2019; Carvalho et al., 2023; Rodriguez-Sanchez & Konidaris, 2024) or learning policies and value functions at multiple levels of abstraction (e.g., Precup & Sutton, 1997; Precup et al., 1997; 1998; Dayan & Hinton, 1992; Dietterich, 1998; Vezhnevets et al., 2017; Kulkarni et al., 2016; Gürtler et al., 2021; Nachum et al., 2018; Levy et al., 2019; Park et al., 2023; 2025b). Our compositional planning method can be viewed both as planning over options with a simple termination condition – where each policy terminates after a random number of steps – and as a hierarchical RL method that replaces the high-level policy with a test-time planning procedure.

# B. Algorithms

We present the full TD-HC algorithm along with COMPPLAN and the two proposal distributions outlined herein.

---

**Algorithm 1** Temporal Difference Flows with Horizon Consistency

---

1: **Inputs**: offline dataset $\mathcal{D}$, policy $\pi$, batch size $K$, Polyak coefficient $\zeta$, randomly initialized weights $\theta$, learning rate $\eta$, maximum discount factor $\gamma_{\max} \in [0,1)$, horizon consistency proportion $\tau_c \in [0,1]$.

2: **for** $n = 1, \ldots$ **do**

3:     Sample mini-batch $\{(S_k, A_k, S_k', A_k')\}_{k=1}^K$ from $\mathcal{D}$

4:     **for** $k = 1, \ldots, K$ **do**

5:         $t_k \sim \mathcal{U}([0,1])$

6:         $\gamma_k \sim \mathcal{U}([0, \gamma_{\max}])$

7:

8:         #`One-Step Term`

9:         $X_0 \sim p_0(\cdot)$

10:         $\vec{X}_{t_k} \leftarrow (1 - t_k)X_0 + t_k S_k'$

11:         $\vec{\ell}_k(\theta) = \left\| v_{t_k}(\vec{X}_{t_k} \mid S_k, A_k, \gamma_k; \theta) - (S_k' - X_0) \right\|^2$

12:

13:         **if** $k \leq \lceil K \cdot \tau_c \rceil$ **then**

14:             $\beta_k \sim \mathcal{U}([0, \gamma_k])$

15:             #`β-Bootstrap Term`

16:             $X_0 \sim p_0(\cdot)$

17:             $\widehat{X}_{t_k}^{\beta} \leftarrow \psi_{t_k}(X_0 \mid S_k', A_k', \beta_k; \bar{\theta})$

18:             $\widehat{\ell}_k^{\beta}(\theta) = \left\| v_{t_k}(\widehat{X}_{t_k}^{\beta} \mid S_k, A_k, \gamma_k; \theta) - v_{t_k}(\widehat{X}_{t_k}^{\beta} \mid S_k', A_k', \beta_k; \bar{\theta}) \right\|^2$

19:             #`γ-Bootstrap Term`

20:             $(X_0, X_0'') \sim p_0(\cdot)$

21:             $S_k'' \leftarrow \psi_1(X_0'' \mid S_k', A_k', \beta_k; \bar{\theta})$

22:             $A_k'' \sim \pi(\cdot \mid S_k'')$

23:             $\widehat{X}_{t_k}^{\gamma} \leftarrow \psi_{t_k}(X_0 \mid S_k'', A_k'', \gamma_k; \bar{\theta})$

24:             $\widehat{\ell}_k^{\gamma}(\theta) = \left\| v_{t_k}(\widehat{X}_{t_k}^{\gamma} \mid S_k, A_k, \gamma_k; \theta) - v_{t_k}(\widehat{X}_{t_k}^{\gamma} \mid S_k'', A_k'', \gamma_k; \bar{\theta}) \right\|^2$

25:             #`Mixture Loss`

26:             $\ell_k(\theta) = (1 - \gamma_k)\vec{\ell}_k(\theta) + \gamma_k \frac{1 - \gamma_k}{1 - \beta_k}\widehat{\ell}_k^{\beta}(\theta) + \gamma_k \frac{\gamma_k - \beta_k}{1 - \beta_k}\widehat{\ell}_k^{\gamma}(\theta)$

27:         **else**

28:             #`γ-Bootstrap Term`

29:             $X_0 \sim p_0(\cdot)$

30:             $\widehat{X}_{t_k}^{\gamma} \leftarrow \psi_{t_k}(X_0 \mid S_k', A_k', \gamma_k; \bar{\theta})$

31:             $\widehat{\ell}_k^{\gamma}(\theta) = \left\| v_{t_k}(\widehat{X}_{t_k}^{\gamma} \mid S_k, A_k, \gamma_k; \theta) - v_{t_k}(\widehat{X}_{t_k}^{\gamma} \mid S_k', A_k', \gamma_k; \bar{\theta}) \right\|^2$

32:             #`Mixture Loss`

33:             $\ell_k(\theta) = (1 - \gamma_k)\vec{\ell}_k(\theta) + \gamma_k \widehat{\ell}_k^{\gamma}(\theta)$

34:         **end if**

35:     **end for**

36:     #`Compute loss`

37:     $\ell(\theta) = \frac{1}{K}\sum_{k=1}^K \ell_k(\theta)$

38:     #`Perform gradient step`

39:     $\theta \leftarrow \theta - \eta \nabla_\theta \ell(\theta)$

40:     #`Update parameters of target vector field`

41:     $\bar{\theta} \leftarrow \zeta\bar{\theta} + (1 - \zeta)\theta$

42: **end for**

---

---

**Algorithm 2** Compositional Planning with Jumpy World Models

---

1: **Inputs**: parameterized class of policies $\{\pi_z\}_{z\in\mathsf{Z}}$, geometric horizon model $m_\gamma^{\pi_z}$, policy sequence length $K$, proposal distribution
   $\rho : \mathsf{S} \to \mathscr{P}(\mathsf{Z}^K)$, number of proposals $M$, number of monte-carlo samples $N$, reward function $r$, effective discount factors $\{\beta_k\}_{k=1}^K$,
   mixture weights $\{w_k\}_{k=1}^K$
2: **function** COMPPLAN($s$)
3:    **for** $i = 1, \ldots, M$ **do**
4:        `#Sample policy sequence from proposal distribution`
5:        $(z_1^{(i)}, \ldots, z_K^{(i)}) \sim \rho(\cdot \mid s)$
6:
7:        `#Sample initial action`
8:        $a_1^{(i)} \sim \pi_{z_1^{(i)}}(\cdot \mid s)$
9:
10:       `#Monte Carlo Q-value estimation (Lemma 1)`
11:      **for** $j = 1, \ldots, N$ **do**
12:         $(S_0, A_0) \leftarrow (s, a_1^{(i)})$
13:         **for** $k = 1, \ldots, K$ **do**
14:           $S_k \sim m_{\beta_k}^{\pi_{z_k^{(i)}}}(\cdot \mid S_{k-1}, A_{k-1})$
15:           $A_k \sim \pi_{z_{k+1}^{(i)}}(\cdot \mid S_k)$
16:         **end for**
17:         $\widehat{Q}^{(i,j)} \leftarrow (1-\gamma)^{-1} \sum_{k=1}^K w_k \cdot r(S_k)$
18:      **end for**
19:      $\widehat{Q}^{(i)} \leftarrow \frac{1}{N} \sum_{j=1}^N \widehat{Q}^{(i,j)}$
20:    **end for**
21:
22:    `#Select best candidate`
23:    $i^* \leftarrow \arg\max_{i \in [\![M]\!]} \widehat{Q}^{(i)}$
24:
25:    `#Return optimal action and policy`
26:    **return** $(a_1^{(i^*)}, z_1^{(i^*)})$
27: **end function**

---

---

**Algorithm 3** Goal-Conditioned Proposal

---

1: **Inputs**: geometric horizon model $m_\gamma^{\pi_z}$, policy sequence
   length $K$, effective discount factors $\{\beta_k\}_{k=1}^K$
2: **function** GOALCONDPROPOSAL($s, g$)
3:    `#Chain GHM samples toward goal`
4:    $z_0 \leftarrow s$
5:    **for** $k = 1, \ldots, K$ **do**
6:        $A_{k-1} \sim \pi_g(\cdot \mid z_{k-1})$
7:        $z_k \sim m_{\beta_k}^{\pi_g}(\cdot \mid z_{k-1}, A_{k-1})$
8:    **end for**
9:    **return** $(z_1, \ldots, z_K)$
10: **end function**

---

**Algorithm 4** Unconditional Proposal

---

1: **Inputs**: unconditional (behavior policy $\mu$) geometric hori-
   zon model $m_\gamma^\mu$, policy sequence length $K$, effective discount
   factors $\{\beta_k\}_{k=1}^K$
2: **function** UNCONDPROPOSAL($s$)
3:    `#Chain unconditional GHM samples`
4:    $z_0 \leftarrow s$
5:    **for** $k = 1, \ldots, K$ **do**
6:        $z_k \sim m_{\beta_k}^\mu(\cdot \mid z_{k-1})$
7:    **end for**
8:    **return** $(z_1, \ldots, z_K)$
9: **end function**

---

*Figure 2.* Two proposal distributions for goal-conditioned compositional planning. **Left:** GOALCONDPROPOSAL samples subgoal sequences by chaining GHM predictions conditioned on the goal $g \in \mathsf{S}$, guiding the agent toward the target. **Right:** Unconditional proposal samples from the behavior policy's GHM that can be trained alongside $m_\gamma^{\pi_z}$ by periodically setting $z = \varnothing, a = \varnothing$.

## C. Theoretical Results

**Theorem 1.** *Let $\nu := \pi_{z_1} \xrightarrow{\alpha_1} \pi_{z_2} \cdots \xrightarrow{\alpha_{n-1}} \pi_{z_n}$ be a geometric switching policy with global discount factor $\gamma \in (0, 1)$, effective discount factors $\{\beta_k\}_{k=1}^n$, and weights $\{w_k\}_{k=1}^n$ from Definition 1. For any state-action pair $(s, a)$, the successor measure of $\nu$ decomposes as:*

$$m_\gamma^\nu(\mathrm{d}s^+ \mid s, a) = \sum_{k=1}^n w_k \int_{\substack{s_1,\ldots,s_{k-1} \\ a_1,\ldots,a_{k-1}}} m_{\beta_1}^{\pi_{z_1}}(\mathrm{d}s_1 \mid s, a)\pi_{z_2}(\mathrm{d}a_1 \mid s_1) \cdots m_{\beta_k}^{\pi_{z_k}}(\mathrm{d}s^+ \mid s_{k-1}, a_{k-1}).$$

*Proof.* Let's denote $\nu_{l:n} = \pi_{z_l} \xrightarrow{\alpha_l} \pi_{z_{l+1}} \cdots \xrightarrow{\alpha_{n-1}} \pi_{z_n}$ the geometric switching policy that starts by $\pi_{z_l}$. we will proceed by induction over $l \in \{n, n-1, \ldots 1\}$ to show that:

$$m_\gamma^{\nu_{l:n}}(\mathrm{d}s' \mid s, a) = \sum_{k=l}^n \frac{1-\gamma}{1-\beta_k}\left(\prod_{i=l}^{k-1} \frac{\gamma-\beta_i}{1-\beta_i}\right) \int_{\substack{s_l,\ldots,s_{k-1} \\ a_l,\ldots,a_{k-1}}} m_{\beta_l}^{\pi_{z_l}}(\mathrm{d}s_l \mid s, a)\pi_{z_{l+1}}(\mathrm{d}a_l \mid s_l) \ldots m_{\beta_k}^{\pi_{z_k}}(\mathrm{d}s' \mid s_{k-1}, a_{k-1}),$$

(8)

where $(s_{l-1}, a_{l-1}) = (s, a)$.

For the case $l = n$, it is straightforward to see that the induction hypothesis (8) is satisfied since $m_\gamma^{\nu_{n:n}} = m_\gamma^{\pi_{z_n}}$.

Let us now assume that the induction hypothesis (8) holds for $l+1 \in \{n, n-1, \ldots, 2\}$. Our goal is to demonstrate that it also holds for $l$.

After executing a single step of $\nu_{l:n}$, two outcomes are possible:: with probability $(1 - \alpha_l)$, we remain committed to $\nu_{l:n}$, or with probability $\alpha_l$, we switch to the next policy $\pi_{z_{l+1}}$, thereby continuing the episode with $\nu_{l+1:n}$. This leads to the following Bellman-equation:

$$m^{\nu_{l:n}} = (1-\gamma)P + \gamma(1-\alpha_l)P^{\pi_{z_l}}m^{\nu_{l:n}} + \gamma\alpha_l P^{\pi_{z_{l+1}}}m^{\nu_{l+1:n}}$$

which implies

$$(I - \gamma\beta_l P^{\pi_{z_l}})m^{\nu_{l:n}} = (1-\gamma)P + \gamma\alpha_l P^{\pi_{z_{l+1}}}m^{\nu_{l+1:n}}$$
$$\implies m^{\nu_{l:n}} = (1-\gamma)(I - \gamma\beta_l P^{\pi_{z_l}})^{-1}P + \gamma\alpha_l(I - \gamma\beta_l P^{\pi_{z_l}})^{-1}P^{\pi_{z_{l+1}}}m^{\nu_{l+1:n}}$$
$$\implies m^{\nu_{l:n}}(\mathrm{d}s' \mid s, a) = \frac{1-\gamma}{1-\beta_l}m_{\beta_l}^{\pi_l}(\mathrm{d}s' \mid s, a) + \frac{\gamma-\beta_l}{1-\beta_l}\int_{s_l} m_{\beta_l}^{\pi_l}(\mathrm{d}s_l \mid s, a)\pi_{\pi_{z_{l+1}}}(\mathrm{d}a_l \mid s_l)m^{\nu_{l+1:n}}(\mathrm{d}s' \mid s_l, a_l).$$

Using the induction hypothesis for $l + 1$, we find:

$$m^{\nu_{l:n}}(\mathrm{d}s' \mid s, a)$$
$$= \frac{1-\gamma}{1-\beta_l}m_{\beta_l}^{\pi_l}(\mathrm{d}s' \mid s, a)$$
$$+ \frac{\gamma-\beta_l}{1-\beta_l}\int_{s_l} m_{\beta_l}^{\pi_l}(\mathrm{d}s_l \mid s, a)\pi_{\pi_{z_{l+1}}}(\mathrm{d}a_l \mid s_l)$$
$$\times \sum_{k=l+1}^n \frac{1-\gamma}{1-\beta_k}\left(\prod_{i=l+1}^{k-1} \frac{\gamma-\beta_i}{1-\beta_i}\right)\int_{\substack{s_{l+1},\ldots,s_{k-1} \\ a_{l+1},\ldots,a_{k-1}}} m_{\beta_{l+1}}^{\pi_{z_{l+1}}}(\mathrm{d}s_{l+1} \mid s, a)\pi_{z_{l+2}}(\mathrm{d}a_{l+1} \mid s_{l+1}) \ldots m_{\beta_k}^{\pi_{z_k}}(\mathrm{d}s' \mid s_{k-1}, a_{k-1})$$
$$= \frac{1-\gamma}{1-\beta_l}m_{\beta_l}^{\pi_l}(\mathrm{d}s' \mid s, a)$$
$$+ \sum_{k=l+1}^n \frac{1-\gamma}{1-\beta_k}\left(\prod_{i=l}^{k-1} \frac{\gamma-\beta_i}{1-\beta_i}\right)\int_{\substack{s_l,\ldots,s_{k-1} \\ a_l,\ldots,a_{k-1}}} m_{\beta_l}^{\pi_{z_l}}(\mathrm{d}s_l \mid s, a)\pi_{z_{l+1}}(\mathrm{d}a_l \mid s_l) \ldots m_{\beta_k}^{\pi_{z_k}}(\mathrm{d}s' \mid s_{k-1}, a_{k-1})$$
$$= \sum_{k=l}^n \frac{1-\gamma}{1-\beta_k}\left(\prod_{i=l}^{k-1} \frac{\gamma-\beta_i}{1-\beta_i}\right)\int_{\substack{s_l,\ldots,s_{k-1} \\ a_l,\ldots,a_{k-1}}} m_{\beta_l}^{\pi_{z_l}}(\mathrm{d}s_l \mid s, a)\pi_{z_{l+1}}(\mathrm{d}a_l \mid s_l) \ldots m_{\beta_k}^{\pi_{z_k}}(\mathrm{d}s' \mid s_{k-1}, a_{k-1})$$

which shows the desired result. $\square$

### C.1. Multi-Timescale Temporal Difference Flows with Horizon Consistency

The results in this section generalize those found in Farebrother et al. (2025) to arbitrary mixture distributions.

**Lemma 2.** *Let $\{v_t^i\}_{i\in[\![N]\!]}$ a family of $N \in \mathbb{N}$ vector fields that generate the probability paths $\{p_t^i\}_{i\in[\![N]\!]}$. Then, the mixture probability path $p_t = \sum_i \lambda_i p_t^i$, where $\{\lambda_i\}_{i\in[\![N]\!]} \in [0,1]$ and $\sum_i \lambda_i = 1$ is generated by the vector field*

$$v_t := \frac{\sum_i \lambda_i p_t^i v_t^i}{\sum_i \lambda_i p_t^i}. \tag{9}$$

*Proof.* Since $v_i^t$ generates $p_t^i$, we know from the continuity equation that:

$$\forall i \in [\![N]\!], \frac{\partial p_t^i}{\partial t} = \mathrm{div}(p_t^i v_t^i)$$

where div denotes the divergence operator. Then, by linearity of div,

$$\begin{aligned}
\frac{\partial p_t}{\partial t} &= \frac{\partial \left(\sum_i \lambda_i p_t^i\right)}{\partial t} \\
&= \sum_i \lambda_i \mathrm{div}(p_t^i v_t^i) \\
&= \mathrm{div}\left(\sum_i \lambda_i p_t^i v_t^i\right) \\
&= \mathrm{div}\left(\frac{\sum_i \lambda_i p_t^i v_t^i}{\sum_i \lambda_i p_t^i} \sum_i \lambda_i p_t^i\right) \\
&= \mathrm{div}(v_t p_t).
\end{aligned}$$

Hence, $(v_t, p_t)$ satisfies the continuity equation, which implies that $v_t$ generates $p_t$. $\qquad\square$

**Lemma 3.** *Let $\{v_t^i\}_{i\in[\![N]\!]}$ a family of $N \in \mathbb{N}$ vector fields that generate the probability paths $\{p_t^i\}_{i\in[\![N]\!]}$. For $\lambda_i \in [0,1]$ such that $\sum_i \lambda_i = 1$, the vector field $v_t = \frac{\sum_i \lambda_i p_t^i v_t^i}{\sum_i \lambda_i p_t^i}$ satisfies*

$$v_t = \underset{v:\mathbb{R}^d \to \mathbb{R}^d}{\arg\min} \left\{ \sum_i \lambda_i \mathbb{E}_{x_t \sim p_t^1} \left[ \|v_t(x_t) - v_t^i(x_t)\|^2 \right] \right\}.$$

*Proof.* Let $\ell_t(v) := \sum_i \lambda_i \mathbb{E}_{x_t \sim p_t^i} \left[ \|v_t(x_t) - v_t^i(x_t)\|^2 \right]$. The functional derivative of this quantity wrt $v$ evaluated at some point $x$ is

$$\nabla_v \ell_t(v)(x) = \sum_i \lambda_i p_i^t(x)(v_t(x) - v_t^i(x)).$$

Setting this to zero and solving for $v_t(x)$ yields the result. $\qquad\square$

The consistency operator in equation (5) combines three different distributions. Lemmas 2 and 3 indicate that we can construct distinct probability paths for each distribution as follows

1. For the first distribution *i.e* $P(\cdot \mid s,a)$, We apply the standard Conditional Flow Matching (CFM) approach, where the probability path is defined as the marginal over a simple conditional path (specifically, we use the Optimal Transport (OT) path):

$$q_t(x|s,a) = \mathbb{E}_{S' \sim P(\cdot|s,a)} \left[ \mathcal{N}(x; tS', (1-t)^2) \right] \tag{10}$$

where $\mathcal{N}(x; tS', (1-t)^2)$ is the gaussian distribution of mean $tS'$ and variance $(1-t)^2$. This leads to the standard CFM objective:

$$\mathbb{E}_{\substack{t,(S,A,S') \\ X_0 \sim p_0, X_t = tS' + (1-t)X_0}} \left[ \left\| v_t(X_t \mid S, A; \theta, \gamma) - (S' - X_0) \right\|^2 \right] \tag{11}$$

2. For the second distribution, *i.e.,* $\mathbb{E}_{S'\sim P(\cdot|s,a),A'\sim\pi(\cdot|S')}\left[m_\beta^\pi(\cdot\mid S',A')\right]=(P^\pi m_\beta^\pi)(\cdot\mid s,a)$, we leverage that $m_\beta^\pi$ is parametrized by a flow matching model to define the probability path:

$$q_t(x\mid s,a)=\mathbb{E}_{S'\sim P(\cdot|s,a),A'\sim\pi(\cdot|S')}\left[p_{0\#}\psi_t(\cdot\mid S',A';\theta,\beta)\right] \tag{12}$$

$q_t$ is a valid probability path, satisfying the boundary conditions: $q_0(x\mid s,a)=\mathbb{E}_{S'\sim P(\cdot|s,a),A'\sim\pi(\cdot|S')}\left[p_0(x)\right]=p_0(x)$ and $q_1=P^\pi m_\beta^\pi$. $q_t$ can be interpreted as aggregation of conditional paths $p_{0\#}\psi_t(\cdot\mid S',A',\beta,\theta)$, for which we have access to their vector field $v_t(X_t\mid S',A';\theta,\beta)$. Using the equivalence between marginal flow matching and conditional flow matching (Lipman et al., 2023), we arrive at the following objective.

$$\mathbb{E}_{\substack{t,(S,A,S'),A'\sim\pi(\cdot|S')\\X_t\sim p_{0\#}\psi_t(\cdot|S',A',\beta;\bar\theta)}}\left[\left\|v_t(X_t\mid S,A,\gamma;\theta)-v_t(X_t\mid S',A',\beta;\bar\theta)\right\|^2\right] \tag{13}$$

3. Similarly, for the third distribution $\mathbb{E}_{\substack{S'\sim P(\cdot|s,a),A'\sim\pi(\cdot|S')\\S''\sim m_\beta^\pi(\cdot|S',A'),A''\sim\pi(\cdot|S'')}}\left[m_\gamma^\pi(\cdot\mid S'',A'')\right]$, we again leverage that $m_\gamma^\pi$ is parametrized by flow matching model to define the following probability path:

$$q_t(x\mid s,a)=\mathbb{E}_{\substack{S'\sim P(\cdot|s,a),A'\sim\pi(\cdot|S')\\S''\sim p_{0\#}\psi_1(\cdot|S',A',\beta;\bar\theta),A''\sim\pi(\cdot|S'')}}\left[p_{0\#}\psi_t(\cdot\mid S',A';\theta,\gamma)\right] \tag{14}$$

$q_t$ can be interpreted as aggregation of conditional paths $p_{0\#}\psi_t(\cdot\mid S',A',\gamma,\theta)$, for which we have access to their vector field $v_t(X_t\mid S',A',\gamma;\theta)$. Using the equivalence between marginal flow matching and conditional flow matching (Lipman et al., 2023), we arrive at the following objective.

$$\mathbb{E}_{\substack{t,(S,A,S'),A'\sim\pi(\cdot|S')\\S''\sim p_{0\#}\psi_1(\cdot|S',A',\beta;\bar\theta)\\X_t\sim p_{0\#}\psi_t(\cdot|S'',A'',\gamma;\bar\theta)}}\left[\left\|v_t(X_t\mid S,A,\gamma;\theta)-v_t(X_t\mid S'',A'',\gamma;\bar\theta)\right\|^2\right] \tag{15}$$

# D. Additional Results

## D.1. Compositional Planning / Zero-Shot Results

We report the full planning results here. Table 4 summarizes the success rate for each task. We evaluate 7 domains with 5 tasks per domain, for a total of 35 tasks. For each task we evaluate the base policy and COMPPLAN by rolling out 10 trajectories. We report standard deviation over the 3 seeds used for GHM training, for the base policy we do not have multiple seeds. We see that COMPPLAN is better than the base policy in almost all the tasks. Figure 3 shows a bar plot comparing the domain-averaged success rates of COMPPLAN and the zero-shot baseline.

*Table 4.* Success rate (↑) per task for base policies $\pi_g$ (Zero Shot) and COMPPLAN. Mean and standard deviation reported over 3 seeds. Blue and red denote an increase and decrease w.r.t. zero-shot with gray indicating no significant difference.

| Domain | Task | CRL | | GC-1S | | GC-BC | | GC-TD3 | | HFBC | |
|---|---|---|---|---|---|---|---|---|---|---|---|
| | | Zero Shot | COMPPLAN | Zero Shot | COMPPLAN | Zero Shot | COMPPLAN | Zero Shot | COMPPLAN | Zero Shot | COMPPLAN |
| ANTMAZE-MEDIUM | 1 | 0.950 | 0.970 (0.030) | 0.400 | 0.930 (0.070) | 0.400 | 0.870 (0.090) | 0.650 | 0.800 (0.060) | 0.900 | 0.870 (0.030) |
| | 2 | 0.950 | 0.900 (0.100) | 0.800 | 1.000 (0.000) | 0.550 | 0.830 (0.090) | 0.900 | 0.530 (0.120) | 0.900 | 0.970 (0.030) |
| | 3 | 0.650 | 0.970 (0.030) | 0.100 | 0.500 (0.120) | 0.600 | 0.900 (0.060) | 0.450 | 0.770 (0.030) | 1.000 | 0.970 (0.030) |
| | 4 | 0.900 | 1.000 (0.000) | 0.650 | 0.930 (0.030) | 0.100 | 0.770 (0.090) | 0.550 | 0.600 (0.100) | 1.000 | 0.970 (0.030) |
| | 5 | 0.950 | 1.000 (0.000) | 0.850 | 0.970 (0.030) | 0.800 | 0.870 (0.090) | 0.700 | 0.570 (0.070) | 0.900 | 0.930 (0.030) |
| ANTMAZE-LARGE | 1 | 0.800 | 0.830 (0.030) | 0.100 | 0.770 (0.070) | 0.150 | 0.570 (0.070) | 0.300 | 0.500 (0.120) | 0.800 | 0.970 (0.030) |
| | 2 | 0.600 | 0.770 (0.030) | 0.150 | 0.630 (0.120) | 0.100 | 0.730 (0.030) | 0.100 | 0.530 (0.090) | 0.600 | 0.800 (0.000) |
| | 3 | 0.850 | 0.930 (0.030) | 0.800 | 0.830 (0.090) | 0.550 | 0.900 (0.060) | 0.750 | 0.500 (0.100) | 1.000 | 0.970 (0.030) |
| | 4 | 0.950 | 1.000 (0.000) | 0.000 | 0.430 (0.030) | 0.000 | 0.670 (0.090) | 0.000 | 0.400 (0.060) | 0.900 | 0.900 (0.060) |
| | 5 | 1.000 | 0.970 (0.030) | 0.000 | 0.370 (0.120) | 0.100 | 0.770 (0.090) | 0.000 | 0.470 (0.090) | 0.600 | 0.970 (0.030) |
| ANTMAZE-GIANT | 1 | 0.000 | 0.070 (0.030) | 0.000 | 0.000 (0.000) | 0.000 | 0.000 (0.000) | 0.000 | 0.000 (0.000) | 0.400 | 0.730 (0.090) |
| | 2 | 0.000 | 0.670 (0.070) | 0.000 | 0.000 (0.000) | 0.000 | 0.000 (0.000) | 0.000 | 0.030 (0.030) | 0.300 | 0.830 (0.030) |
| | 3 | 0.000 | 0.100 (0.060) | 0.000 | 0.000 (0.000) | 0.000 | 0.000 (0.000) | 0.000 | 0.000 (0.000) | 0.200 | 0.800 (0.060) |
| | 4 | 0.500 | 0.230 (0.070) | 0.000 | 0.000 (0.000) | 0.000 | 0.000 (0.000) | 0.000 | 0.000 (0.000) | 0.500 | 0.730 (0.030) |
| | 5 | 0.300 | 0.400 (0.120) | 0.000 | 0.100 (0.000) | 0.000 | 0.130 (0.030) | 0.000 | 0.030 (0.030) | 0.700 | 0.830 (0.030) |
| CUBE-1 | 1 | 0.200 | 0.930 (0.030) | 0.300 | 0.470 (0.030) | 1.000 | 1.000 (0.000) | 0.500 | 0.900 (0.060) | 0.800 | 0.970 (0.030) |
| | 2 | 0.100 | 0.830 (0.090) | 0.400 | 0.470 (0.030) | 0.950 | 1.000 (0.000) | 0.850 | 1.000 (0.000) | 0.700 | 1.000 (0.000) |
| | 3 | 0.500 | 0.900 (0.000) | 0.300 | 0.900 (0.060) | 0.950 | 1.000 (0.000) | 0.600 | 0.900 (0.000) | 0.900 | 1.000 (0.000) |
| | 4 | 0.250 | 0.930 (0.030) | 0.500 | 0.770 (0.030) | 1.000 | 1.000 (0.000) | 0.550 | 0.900 (0.060) | 0.700 | 0.970 (0.030) |
| | 5 | 0.350 | 0.700 (0.120) | 0.350 | 0.700 (0.100) | 0.600 | 0.970 (0.030) | 0.400 | 0.870 (0.070) | 0.900 | 0.900 (0.060) |
| CUBE-2 | 1 | 0.100 | 0.870 (0.030) | 0.200 | 0.970 (0.030) | 0.750 | 1.000 (0.000) | 0.300 | 0.970 (0.030) | 1.000 | 1.000 (0.000) |
| | 2 | 0.000 | 0.500 (0.060) | 0.200 | 0.500 (0.170) | 0.000 | 1.000 (0.000) | 0.150 | 0.930 (0.070) | 0.900 | 0.930 (0.030) |
| | 3 | 0.000 | 0.570 (0.030) | 0.050 | 0.600 (0.100) | 0.000 | 1.000 (0.000) | 0.100 | 0.870 (0.030) | 0.900 | 1.000 (0.000) |
| | 4 | 0.000 | 0.070 (0.030) | 0.000 | 0.400 (0.170) | 0.000 | 0.870 (0.070) | 0.000 | 0.430 (0.130) | 0.300 | 0.200 (0.060) |
| | 5 | 0.000 | 0.500 (0.100) | 0.050 | 0.370 (0.070) | 0.000 | 0.970 (0.030) | 0.050 | 0.900 (0.060) | 0.700 | 0.700 (0.100) |
| CUBE-3 | 1 | 0.050 | 1.000 (0.000) | 0.050 | 0.930 (0.030) | 0.450 | 1.000 (0.000) | 0.400 | 1.000 (0.000) | 0.900 | 1.000 (0.000) |
| | 2 | 0.000 | 0.970 (0.030) | 0.000 | 1.000 (0.000) | 0.000 | 1.000 (0.000) | 0.050 | 1.000 (0.000) | 0.900 | 1.000 (0.000) |
| | 3 | 0.000 | 0.930 (0.030) | 0.000 | 0.870 (0.090) | 0.000 | 1.000 (0.000) | 0.150 | 0.930 (0.070) | 0.800 | 0.970 (0.030) |
| | 4 | 0.000 | 0.270 (0.090) | 0.000 | 0.300 (0.060) | 0.000 | 0.700 (0.060) | 0.000 | 0.700 (0.060) | 0.300 | 0.370 (0.120) |
| | 5 | 0.000 | 0.470 (0.030) | 0.000 | 0.230 (0.030) | 0.000 | 0.900 (0.000) | 0.000 | 0.530 (0.170) | 0.300 | 0.800 (0.000) |
| CUBE-4 | 1 | 0.000 | 0.530 (0.070) | 0.050 | 0.970 (0.030) | 0.000 | 1.000 (0.000) | 0.000 | 1.000 (0.000) | 0.600 | 1.000 (0.000) |
| | 2 | 0.000 | 0.870 (0.090) | 0.000 | 0.900 (0.060) | 0.000 | 1.000 (0.000) | 0.000 | 0.970 (0.030) | 0.400 | 0.830 (0.120) |
| | 3 | 0.000 | 0.270 (0.120) | 0.000 | 0.800 (0.000) | 0.000 | 1.000 (0.000) | 0.000 | 0.470 (0.030) | 0.200 | 0.770 (0.030) |
| | 4 | 0.000 | 0.030 (0.030) | 0.000 | 0.100 (0.060) | 0.000 | 0.170 (0.170) | 0.000 | 0.170 (0.120) | 0.000 | 0.200 (0.100) |
| | 5 | 0.000 | 0.230 (0.130) | 0.000 | 0.230 (0.070) | 0.000 | 0.630 (0.130) | 0.000 | 0.230 (0.030) | 0.000 | 0.570 (0.130) |

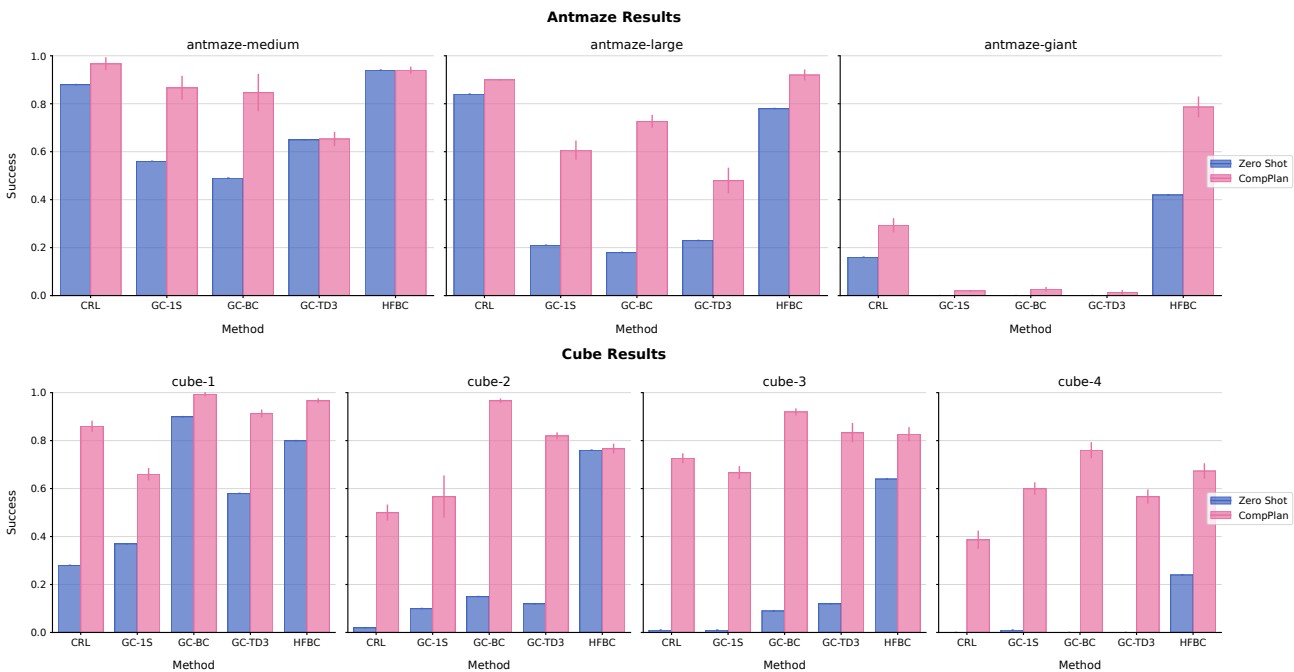

*Figure 3.* Success rate (↑) of base policies $\pi_g$ (Zero Shot) and compositional planning as in Equation (7) with GHMs (COMPPLAN, ours) averaged over tasks.

## D.2. Compositional Planning / GPI / Action Planning Results

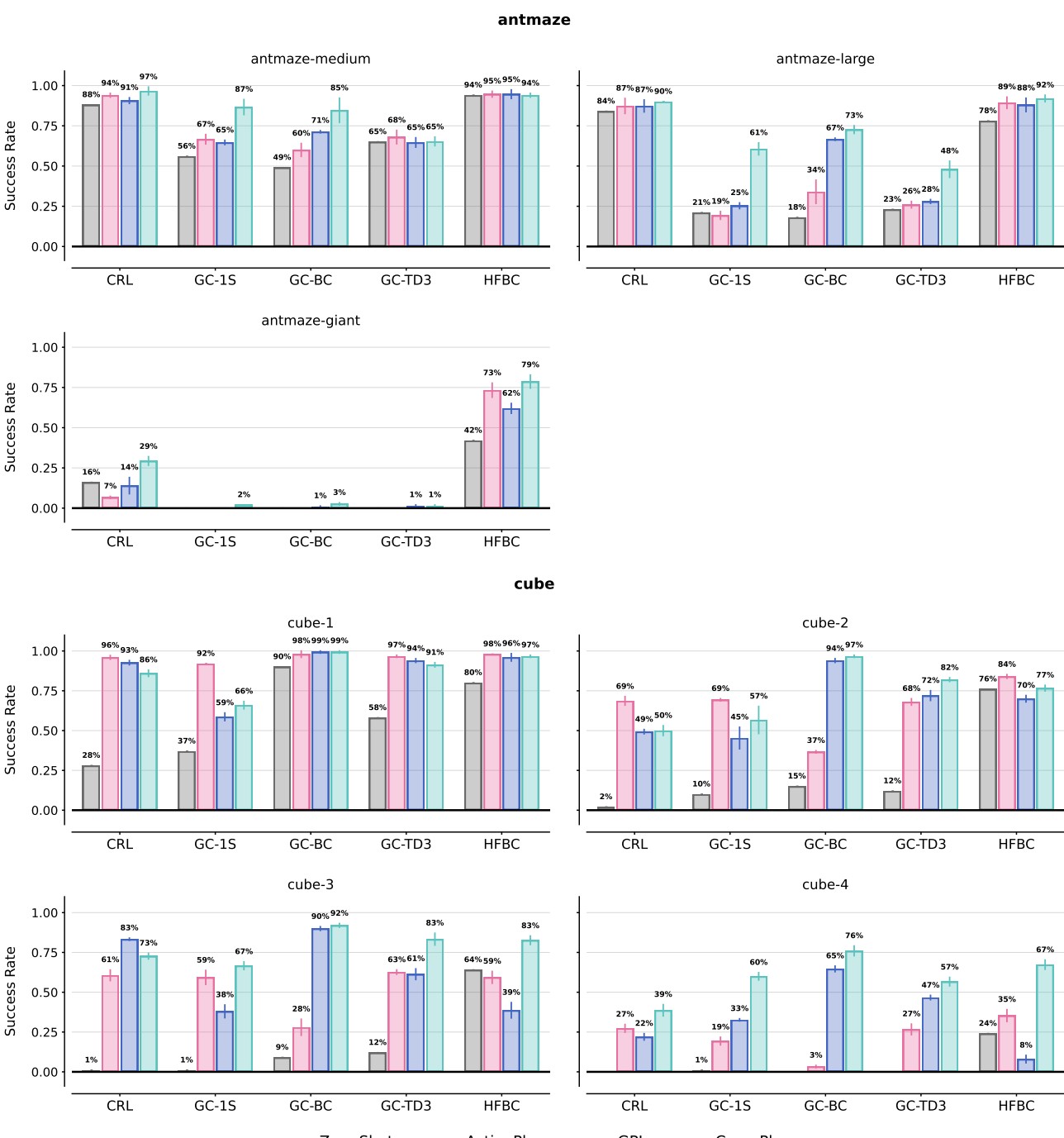

*Figure 4.* Success rate (↑) of planning vs. zero shot. We consider action-level planning (ACTIONPLAN) with a world model; generalized policy improvement (GPI) and compositional planning (COMPPLAN; ours) with GHMs.

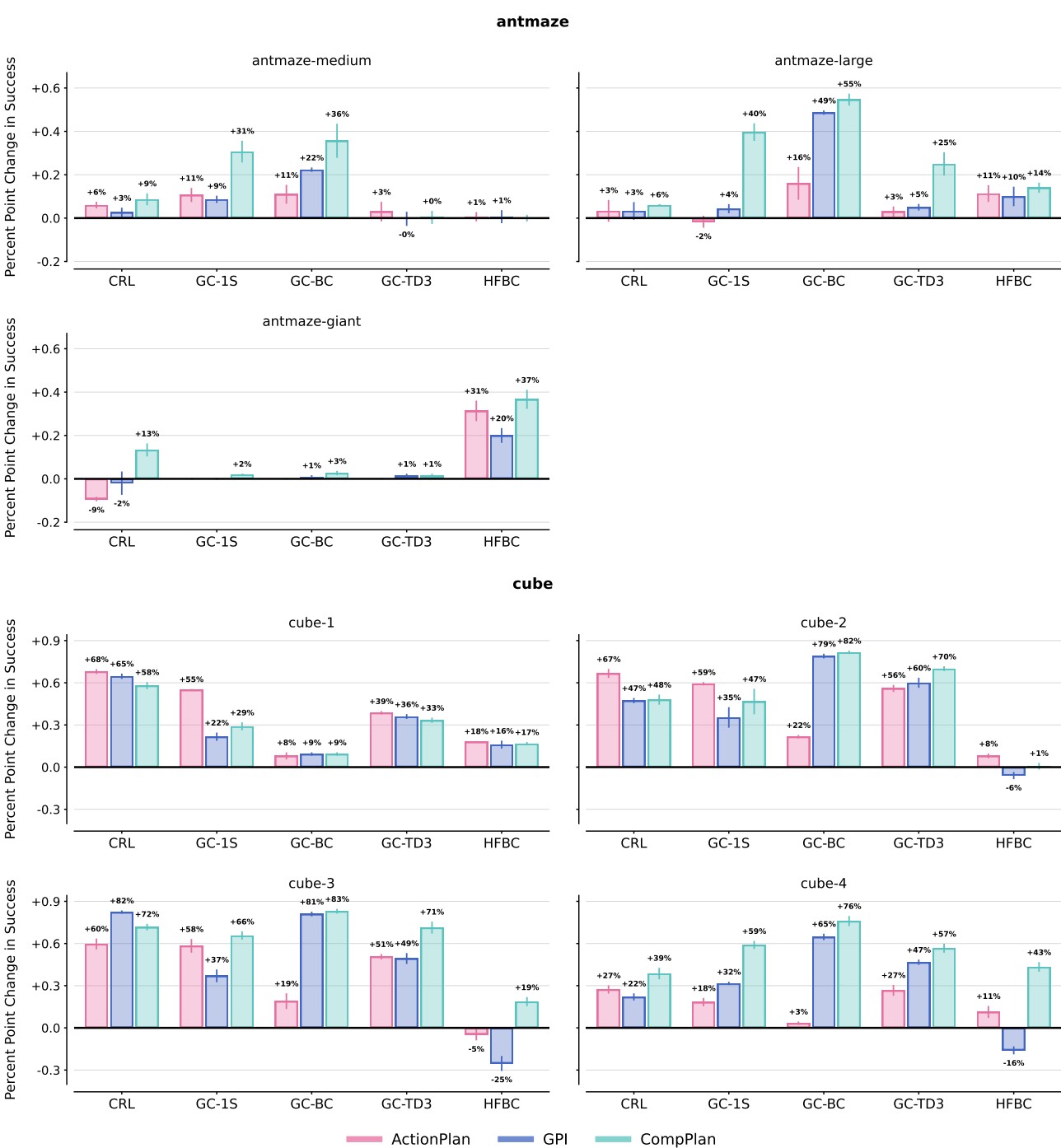

*Figure 5.* Percent point change (↑) of planning over zero shot. We consider action-level planning (ACTIONPLAN) with a world model; generalized policy improvement (GPI) and compositional planning (COMPPLAN; ours) with GHMs.

### D.3. Ablation on the Planning Frequency

In this section, we investigate whether the planning cost can be amortized over time. Specifically, we compare the performance of planning at each time step with planning every N steps. In the latter case, we execute the action maximizing the COMPPLAN objective for the first step and then follow the policy $\pi_{z_1}$ (i.e., the trajectory is $s, a^\star, s_1, \pi_{z_1}(s_1), s_2, \ldots, s_{N-1}, \pi_{z_1}(s_{N-1}), s_N$). Figure 6 reports the average success rate for each domain and method. On average, planning at every step leads to about a 20% improvement compared to planning every 5 steps. This is mostly due to a few cases—most notably, CRL policies in Cube—in which planning every 5 steps results in a large performance drop or complete unlearning. In the majority of experiments, planning every 5 steps does not substantially degrade overall performance, while significantly reducing planning time. Overall, this is a lever that can be used to trade off speed and performance. Finally, Table 5 provides a tabular summary of these results.

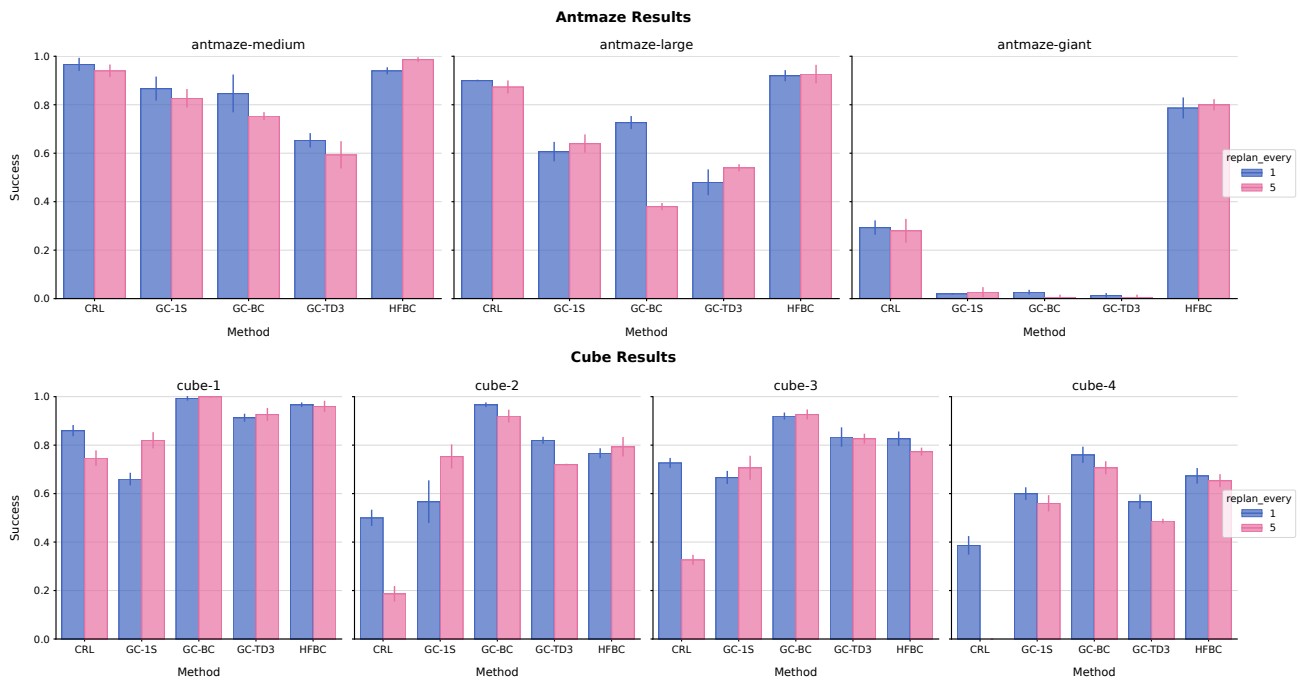

*Figure 6.* Success rate (↑) of COMPPLAN with different base policies when planning at every step of every 5 steps. We report mean and standard deviation 3 seeds for GHM training.

*Table 5.* Success rate (↑) of COMPPLAN with different base policies when replanning every 1 or 5 steps. We report mean and standard deviation over 3 seeds. Best method highlighted in blue; gray indicates no significant difference.

| Domain | CRL | | GC-1S | | GC-BC | | GC-TD3 | | HFBC | |
|---|---|---|---|---|---|---|---|---|---|---|
| | 1 Step | 5 Steps | 1 Step | 5 Steps | 1 Step | 5 Steps | 1 Step | 5 Steps | 1 Step | 5 Steps |
| ANTMAZE-MEDIUM | 0.97 (0.02) | 0.94 (0.02) | 0.87 (0.05) | 0.83 (0.04) | 0.85 (0.08) | 0.75 (0.01) | 0.65 (0.03) | 0.59 (0.05) | 0.94 (0.01) | 0.99 (0.01) |
| ANTMAZE-LARGE | 0.90 (0.00) | 0.87 (0.02) | 0.61 (0.04) | 0.64 (0.03) | 0.73 (0.02) | 0.38 (0.01) | 0.48 (0.05) | 0.54 (0.01) | 0.92 (0.02) | 0.93 (0.04) |
| ANTMAZE-GIANT | 0.29 (0.03) | 0.28 (0.05) | 0.02 (0.00) | 0.03 (0.02) | 0.03 (0.01) | 0.01 (0.01) | 0.01 (0.01) | 0.01 (0.01) | 0.79 (0.04) | 0.80 (0.02) |
| CUBE-1 | 0.86 (0.02) | 0.75 (0.03) | 0.66 (0.02) | 0.82 (0.03) | 0.99 (0.01) | 1.00 (0.00) | 0.91 (0.01) | 0.93 (0.02) | 0.97 (0.01) | 0.96 (0.02) |
| CUBE-2 | 0.50 (0.03) | 0.19 (0.03) | 0.57 (0.09) | 0.75 (0.05) | 0.97 (0.01) | 0.92 (0.02) | 0.82 (0.01) | 0.72 (0.00) | 0.77 (0.02) | 0.79 (0.04) |
| CUBE-3 | 0.73 (0.02) | 0.33 (0.02) | 0.67 (0.02) | 0.71 (0.05) | 0.92 (0.01) | 0.93 (0.02) | 0.83 (0.02) | 0.83 (0.02) | 0.83 (0.03) | 0.77 (0.01) |
| CUBE-4 | 0.39 (0.04) | 0.00 (0.00) | 0.60 (0.02) | 0.56 (0.03) | 0.76 (0.03) | 0.71 (0.02) | 0.57 (0.03) | 0.49 (0.01) | 0.67 (0.03) | 0.65 (0.02) |

## D.4. Ablation on the Planning Objective

As mentioned in the main paper, we can leverage the GHM in several different planning approaches. In this section, we compare two strategies. The first is the approach presented in the paper, in which we optimize both the first action and the policy sequence $(z_1, \ldots, z_n)$:

$$\max_{a_1, z_1, \ldots, z_n} Q_\gamma^{\pi_{z_1} \xrightarrow{\alpha_1} \pi_{z_2} \ldots \xrightarrow{\alpha_{n-1}} \pi_{z_n}} (s, a_1) \tag{16}$$

while the second does not involve action optimization

$$\max_{z_1, \ldots, z_n} V_\gamma^{\pi_{z_1} \xrightarrow{\alpha_1} \pi_{z_2} \ldots \xrightarrow{\alpha_{n-1}} \pi_{z_n}} (s) \tag{17}$$

The difference is that, when using (16), the first action is deterministic, whereas in (17) it is sampled according to $\pi_{z_1}$. Figure 7 shows that maximizing over the first action yields more consistent performance overall, with an average improvement of about 70%.[4] In contrast, planning over the $z$ sequence can fail when the base policy is diffuse (i.e., highly stochastic). This occurs, for example, for CRL policies in the Cube domains and for GC-BC policies in AntMaze. Finally, Table 6 provides a tabular summary of these results.

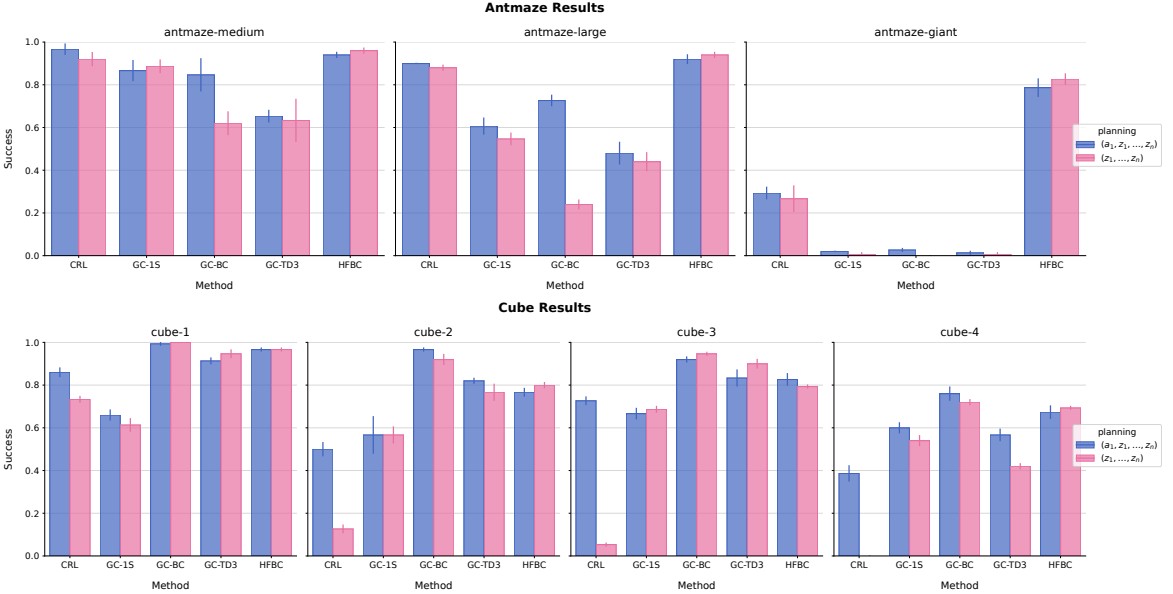

*Figure 7.* Success rate ($\uparrow$) of COMPPLAN with different base policies when maximizing over $(a_1, z_1, \ldots z_n)$ (Eq. 16) or $(z_1, \ldots, z_n)$ (Eq. 17). We report mean and standard deviation 3 seeds for GHM training.

*Table 6.* Success rate ($\uparrow$) of COMPPLAN with (Eq. 16) and without (Eq. 17) action maximization. We report mean and standard deviation over 3 seeds. Best method highlighted in blue; gray indicates no significant difference.

| Domain | CRL | | GC-1S | | GC-BC | | GC-TD3 | | HFBC | |
|---|---|---|---|---|---|---|---|---|---|---|
| | $\max Q$ | $\max V$ | $\max Q$ | $\max V$ | $\max Q$ | $\max V$ | $\max Q$ | $\max V$ | $\max Q$ | $\max V$ |
| ANTMAZE-MEDIUM | 0.97 (0.02) | 0.92 (0.03) | 0.87 (0.05) | 0.89 (0.03) | 0.85 (0.08) | 0.62 (0.05) | 0.65 (0.03) | 0.63 (0.10) | 0.94 (0.01) | 0.96 (0.01) |
| ANTMAZE-LARGE | 0.90 (0.00) | 0.88 (0.01) | 0.61 (0.04) | 0.55 (0.03) | 0.73 (0.02) | 0.24 (0.02) | 0.48 (0.05) | 0.44 (0.04) | 0.92 (0.02) | 0.94 (0.01) |
| ANTMAZE-GIANT | 0.29 (0.03) | 0.27 (0.06) | 0.02 (0.00) | 0.01 (0.01) | 0.03 (0.01) | 0.00 (0.00) | 0.01 (0.01) | 0.01 (0.01) | 0.79 (0.04) | 0.83 (0.02) |
| CUBE-1 | 0.86 (0.02) | 0.73 (0.01) | 0.66 (0.02) | 0.61 (0.03) | 0.99 (0.01) | 1.00 (0.00) | 0.91 (0.01) | 0.95 (0.02) | 0.97 (0.01) | 0.97 (0.01) |
| CUBE-2 | 0.50 (0.03) | 0.13 (0.02) | 0.57 (0.04) | 0.57 (0.04) | 0.97 (0.01) | 0.92 (0.02) | 0.82 (0.01) | 0.77 (0.04) | 0.77 (0.02) | 0.80 (0.01) |
| CUBE-3 | 0.73 (0.02) | 0.05 (0.01) | 0.67 (0.02) | 0.69 (0.01) | 0.92 (0.01) | 0.95 (0.01) | 0.83 (0.04) | 0.90 (0.02) | 0.83 (0.03) | 0.79 (0.01) |
| CUBE-4 | 0.39 (0.04) | 0.00 (0.00) | 0.60 (0.02) | 0.54 (0.02) | 0.76 (0.03) | 0.72 (0.01) | 0.57 (0.03) | 0.42 (0.01) | 0.67 (0.03) | 0.69 (0.01) |

---

[4]This is mostly due to unsuccessful outcomes (i.e., near-zero performance) on certain tasks when Equation 17.

## D.5. Ablation on the Proposal Distribution

In this section, we study the effect of the sampling distribution on the planning procedure. As mentioned in the main paper, GHMs are trained either with the policy condition $z$ or with a learnable token $\varnothing$. In the latter case, the resulting GHM corresponds to the behavioral policy. We compare two planning strategies: (1) sampling from the GHM conditioned on the policy associated with the goal (*conditional proposal*); and (2) sampling from the GHM of the behavioral policy (*unconditional proposal*). Figure 8 shows that, in AntMaze, sampling from the unconditional distribution performs only marginally worse than the conditional proposal, highlighting the robustness of our planning procedure. Table 7 provides a tabular summary of these results.

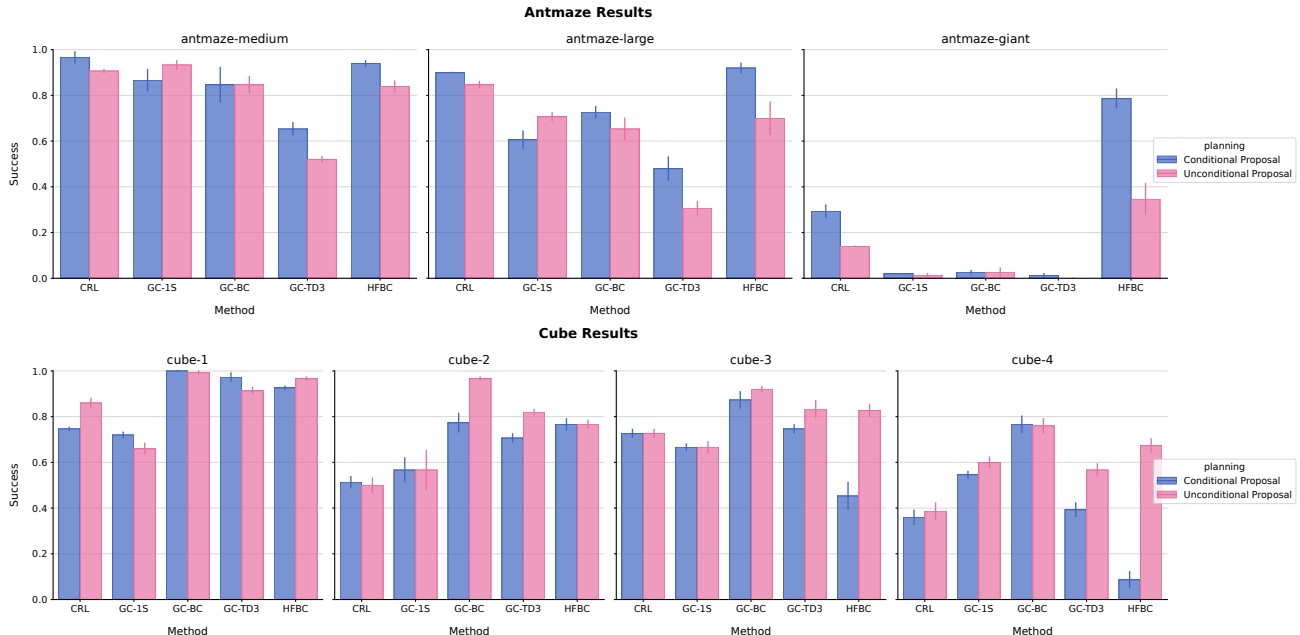

*Figure 8.* Success rate (↑) of COMPPLAN with different base policies when using the conditional or unconditional GHMs to propose subgoals. We report mean and standard deviation 3 seeds for GHM training.

*Table 7.* Success rate (↑) of COMPPLAN with conditional and unconditional proposal. We report mean and standard deviation 3 seeds. Best method highlighted in blue; gray indicates no significant difference.

| Domain | CRL | | GC-1S | | GC-BC | | GC-TD3 | | HFBC | |
|---|---|---|---|---|---|---|---|---|---|---|
| | Cond | Uncond | Cond | Uncond | Cond | Uncond | Cond | Uncond | Cond | Uncond |
| ANTMAZE-MEDIUM | 0.97 (0.02) | 0.91 (0.01) | 0.87 (0.05) | 0.93 (0.02) | 0.85 (0.04) | 0.85 (0.04) | 0.65 (0.03) | 0.52 (0.01) | 0.94 (0.01) | 0.84 (0.02) |
| ANTMAZE-LARGE | 0.90 (0.00) | 0.85 (0.01) | 0.61 (0.04) | 0.71 (0.02) | 0.73 (0.02) | 0.65 (0.05) | 0.48 (0.05) | 0.31 (0.03) | 0.92 (0.02) | 0.70 (0.07) |
| ANTMAZE-GIANT | 0.29 (0.03) | 0.14 (0.00) | 0.02 (0.00) | 0.01 (0.01) | 0.03 (0.02) | 0.03 (0.02) | 0.01 (0.01) | 0.00 (0.00) | 0.79 (0.04) | 0.35 (0.07) |
| CUBE-1 | 0.75 (0.01) | 0.86 (0.02) | 0.72 (0.01) | 0.66 (0.02) | 1.00 (0.00) | 0.99 (0.01) | 0.97 (0.02) | 0.91 (0.01) | 0.93 (0.01) | 0.97 (0.01) |
| CUBE-2 | 0.51 (0.02) | 0.50 (0.03) | 0.57 (0.09) | 0.57 (0.09) | 0.77 (0.04) | 0.97 (0.01) | 0.71 (0.02) | 0.82 (0.01) | 0.77 (0.02) | 0.77 (0.02) |
| CUBE-3 | 0.73 (0.02) | 0.73 (0.02) | 0.67 (0.02) | 0.67 (0.02) | 0.87 (0.04) | 0.92 (0.01) | 0.75 (0.02) | 0.83 (0.04) | 0.45 (0.06) | 0.83 (0.03) |
| CUBE-4 | 0.36 (0.03) | 0.39 (0.04) | 0.55 (0.01) | 0.60 (0.02) | 0.77 (0.04) | 0.76 (0.03) | 0.39 (0.03) | 0.57 (0.03) | 0.09 (0.04) | 0.67 (0.03) |

### D.6. Ablation on the Consistency Objective

This section expands upon §4.4, providing the complete empirical results for the Temporal Difference Horizon Consistency (TD-HC) objective across generative fidelity (Table 8), qualitative predictions (Figure 10), and downstream planning performance (Table 9).

**Generative Fidelity**    Table 8 shows that TD-HC systematically improves the generative accuracy of the TD-FLOW baseline as measured by the Earth Mover's Distance (EMD; Rubner et al., 2000). These gains are especially pronounced in complex domains like ANTMAZE-GIANT where bootstrapping errors easily compound. Figure 10 visually confirms this effect: while both models perform comparably at shorter horizons ($\gamma = 0.99$), TD-FLOW suffers from severe compounding errors at longer horizons ($\gamma = 0.998$), and fails to capture the tail of the distribution. By anchoring its predictions with shorter-horizons, TD-HC successfully respects the topological constraints and properly captures the tail of the true successor state distribution.

**Planning Performance**    Despite the generative advantages of TD-HC at extreme horizons, Table 9 reveals that downstream planning success rates remain broadly similar between the two methods. Because our planning procedure evaluates candidate sequences using moderate effective horizons ($\beta_i \in [0.98, 0.99]$, or roughly $50 - 100$ steps), it does not query the extreme timescales where TD-FLOW breaks down. We expect TD-HC to enable planning over more extreme horizons in the future as benchmarks evolve towards more complex tasks.

*Table 8.* Accuracy (EMD, $\downarrow$) of GHMs trained with (TD-HC) and without (TD-FLOW) our horizon consistency loss (§3.2). Best method is highlighted in blue.

| Domain | CRL | | GC-1S | | GC-BC | | GC-TD3 | | HFBC | |
|---|---|---|---|---|---|---|---|---|---|---|
| | TD-FLOW (✗) | TD-HC (✓) | TD-FLOW (✗) | TD-HC (✓) | TD-FLOW (✗) | TD-HC (✓) | TD-FLOW (✗) | TD-HC (✓) | TD-FLOW (✗) | TD-HC (✓) |
| ANTMAZE-MEDIUM | 4.41 (0.05) | 4.22 (0.06) | 4.40 (0.02) | 4.22 (0.03) | 4.56 (0.05) | 4.36 (0.03) | 4.68 (0.05) | 4.44 (0.05) | 3.38 (0.02) | 3.22 (0.02) |
| ANTMAZE-LARGE | 5.24 (0.07) | 4.81 (0.03) | 5.12 (0.18) | 4.67 (0.04) | 5.32 (0.03) | 4.73 (0.02) | 5.33 (0.07) | 4.83 (0.04) | 3.50 (0.05) | 3.18 (0.01) |
| ANTMAZE-GIANT | 6.77 (0.49) | 5.74 (0.06) | 7.29 (0.69) | 5.25 (0.08) | 6.46 (0.10) | 4.95 (0.12) | 6.51 (0.14) | 5.24 (0.11) | 3.78 (0.03) | 3.00 (0.09) |
| CUBE-1 | 1.60 (0.02) | 1.57 (0.03) | 1.43 (0.00) | 1.33 (0.03) | 1.03 (0.01) | 0.98 (0.01) | 1.12 (0.01) | 1.06 (0.01) | 1.82 (0.06) | 1.27 (0.01) |
| CUBE-2 | 2.36 (0.03) | 2.23 (0.02) | 1.86 (0.04) | 1.71 (0.01) | 1.47 (0.00) | 1.42 (0.01) | 1.89 (0.04) | 1.80 (0.03) | 2.29 (0.09) | 1.54 (0.04) |
| CUBE-3 | 2.15 (0.02) | 2.10 (0.02) | 1.80 (0.04) | 1.71 (0.03) | 1.89 (0.02) | 1.85 (0.02) | 1.91 (0.06) | 1.84 (0.04) | 1.99 (0.09) | 1.55 (0.02) |
| CUBE-4 | 2.41 (0.03) | 2.34 (0.01) | 2.13 (0.03) | 2.05 (0.03) | 2.33 (0.03) | 2.22 (0.02) | 2.21 (0.02) | 2.15 (0.02) | 2.05 (0.05) | 1.61 (0.03) |

*Table 9.* Success rate ($\uparrow$) of COMPPLAN with consistency (TD-HC) and without consistency (TD-FLOW). Mean and standard deviation over 3 seeds. Best method highlighted in blue; gray indicates no significant difference.

| Domain | CRL | | GC-1S | | GC-BC | | GC-TD3 | | HFBC | |
|---|---|---|---|---|---|---|---|---|---|---|
| | TD-FLOW (✗) | TD-HC (✓) | TD-FLOW (✗) | TD-HC (✓) | TD-FLOW (✗) | TD-HC (✓) | TD-FLOW (✗) | TD-HC (✓) | TD-FLOW (✗) | TD-HC (✓) |
| ANTMAZE-MEDIUM | 0.95 (0.01) | 0.97 (0.02) | 0.85 (0.01) | 0.87 (0.05) | 0.91 (0.01) | 0.85 (0.08) | 0.71 (0.03) | 0.65 (0.03) | 0.95 (0.02) | 0.94 (0.01) |
| ANTMAZE-LARGE | 0.91 (0.03) | 0.90 (0.00) | 0.53 (0.09) | 0.61 (0.04) | 0.79 (0.01) | 0.73 (0.02) | 0.51 (0.03) | 0.48 (0.05) | 0.91 (0.03) | 0.92 (0.02) |
| ANTMAZE-GIANT | 0.31 (0.05) | 0.29 (0.03) | 0.01 (0.01) | 0.02 (0.00) | 0.02 (0.00) | 0.03 (0.01) | 0.01 (0.01) | 0.01 (0.01) | 0.81 (0.02) | 0.79 (0.04) |
| CUBE-1 | 0.89 (0.03) | 0.86 (0.02) | 0.55 (0.02) | 0.66 (0.02) | 1.00 (0.00) | 0.99 (0.01) | 0.97 (0.01) | 0.91 (0.01) | 0.97 (0.01) | 0.97 (0.01) |
| CUBE-2 | 0.41 (0.02) | 0.50 (0.03) | 0.57 (0.09) | 0.57 (0.09) | 0.95 (0.02) | 0.97 (0.01) | 0.85 (0.02) | 0.82 (0.01) | 0.84 (0.03) | 0.77 (0.02) |
| CUBE-3 | 0.72 (0.02) | 0.73 (0.02) | 0.72 (0.01) | 0.67 (0.02) | 0.91 (0.02) | 0.92 (0.01) | 0.83 (0.04) | 0.83 (0.04) | 0.83 (0.03) | 0.83 (0.03) |
| CUBE-4 | 0.36 (0.00) | 0.39 (0.04) | 0.56 (0.02) | 0.60 (0.02) | 0.75 (0.02) | 0.76 (0.03) | 0.56 (0.03) | 0.57 (0.03) | 0.64 (0.04) | 0.67 (0.03) |

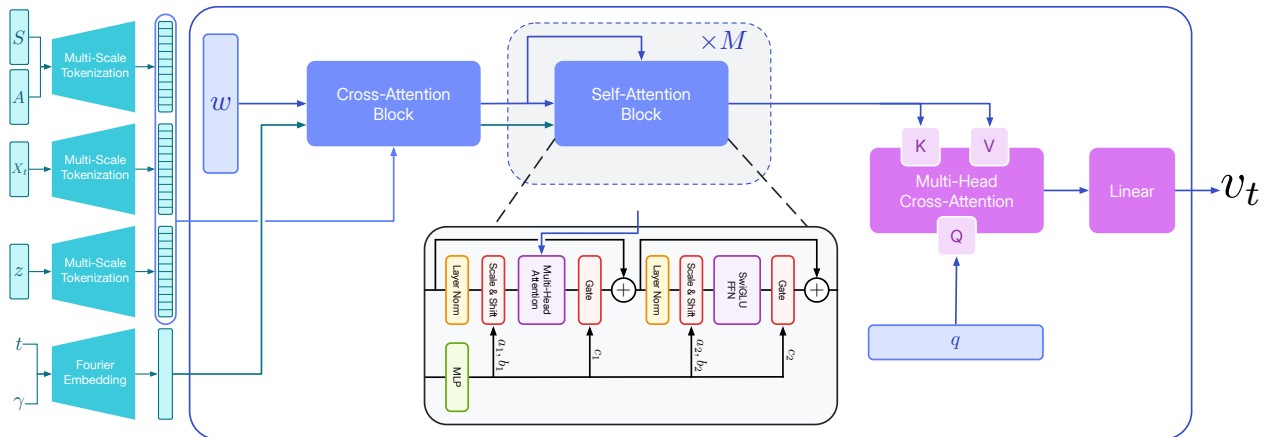

*Figure 9.* Flow perceiver architecture. As input it takes the state and action pair $S, A$, current sample $X_t$, policy embedding $z$, flow timestep $t$, and discount factor $\gamma$. We cross-attend the tokenized input with respect to the latents $w \in \mathbb{R}^{l \times d_{\text{model}}}$ which are further refined by $M$ transformer blocks employing bidirectional self-attention with a SwiGLU FFN. In order to predict the velocity $v_t$ we employ one last cross-attention with respect to another set of learned latents $q \in \mathbb{R}^{d_{\text{model}} \times d_{\text{S}}}$.

### D.7. Scalability to High-Dimensional Environments

To evaluate the scalability of our approach to higher-dimensional continuous control tasks, we conduct an additional set of experiments on the HUMANOIDMAZE domain. Unlike the ANTMAZE tasks, the humanoid agent involves a significantly larger state and action space, presenting a challenge for both policy and GHM learning.

To accommodate this increased dimensionality, we find it beneficial to modify the underlying architecture of our GHM. While the U-Net-style architecture proved highly effective for the standard OGBench tasks, we found it beneficial to adopt a Transformer-based architecture for the humanoid agent. Specifically, we employ a Perceiver-based architecture (Jaegle et al., 2021; 2022) building off of the Perceiver-Actor-Critic (PAC) (Springenberg et al., 2024), which utilizes a combination of self-attention and cross-attention to efficiently process high-dimensional, multi-modal data. In order to deal with the heterogeneous data found in proprioceptive observations we employ their multi-scale tokenization as:

$$\phi(x) = [\tanh(\sigma_1 x), \ldots, \tanh(\sigma_N x)],\qquad(18)$$

for bandwidth values $(\sigma_1, \ldots, \sigma_N)$. Note that we apply $\phi$ per-dimension, so for $s \in \mathbb{R}^d$ we have $\phi(s) \in \mathbb{R}^{N \times d}$ after multi-scale tokenization. Because raw observations exhibit arbitrary scales across different domains, this multi-scale tokenizer represents various magnitudes that the perceiver cross-attended to. A depiction of the architecture can be seen in Figure 9. For the HUMANOIDMAZE experiments we use 12 blocks, 32 latents, hidden dimension 768, FFN expansion factor of 4, 8 bandwidths for multi-scale tokenization, and a per-token linear projection to 256 dimensions before the initial latent cross-attention.

We evaluate COMPPLAN using Hierarchical Flow Behavior Cloning (HFBC) as the base policy class, as it demonstrated the strongest zero-shot performance in our prior evaluations. As shown in Table 10, COMPPLAN consistently and significantly improves upon the base HFBC policies across all maze topologies. Notably, on the most difficult HUMANOIDMAZE-GIANT task, COMPPLAN doubles the success rate of the zero-shot policy, improving success from 0.34 to 0.68.

*Table 10.* Success rate ($\uparrow$) of the base HFBC policy (Zero-Shot) and compositional planning (COMPPLAN; ours) on the HUMANOIDMAZE domain. Best method is highlighted in blue.

| Domain | HFBC | |
|---|---|---|
| | Zero-Shot | COMPPLAN |
| HUMANOIDMAZE-MEDIUM | 0.80 | 0.96 |
| HUMANOIDMAZE-LARGE | 0.40 | 0.64 |
| HUMANOIDMAZE-GIANT | 0.34 | 0.68 |

# E. Qualitative Geometric Horizon Model Samples

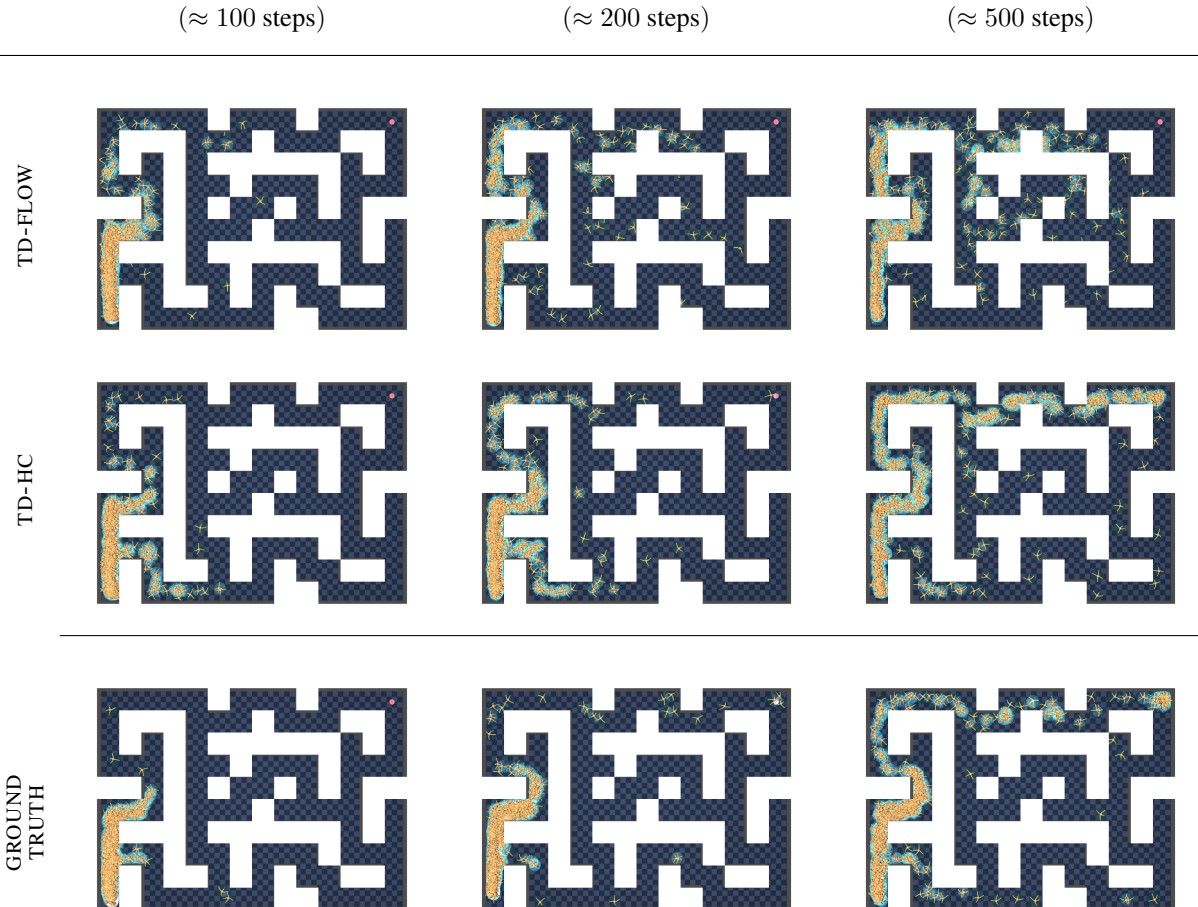

*Figure 10.* Qualitative plots of GHM samples on ANTMAZE-GIANT task 1 at different horizons $(0.99, 0.995, 0.998)$ from TD-FLOW and TD-HC with the last row depicting the ground truth discounted occupancy. As can be seen in the figure, performance is comparable at smaller horizons with TD-HC doing a much better job at capturing the true distribution as the horizon increases.

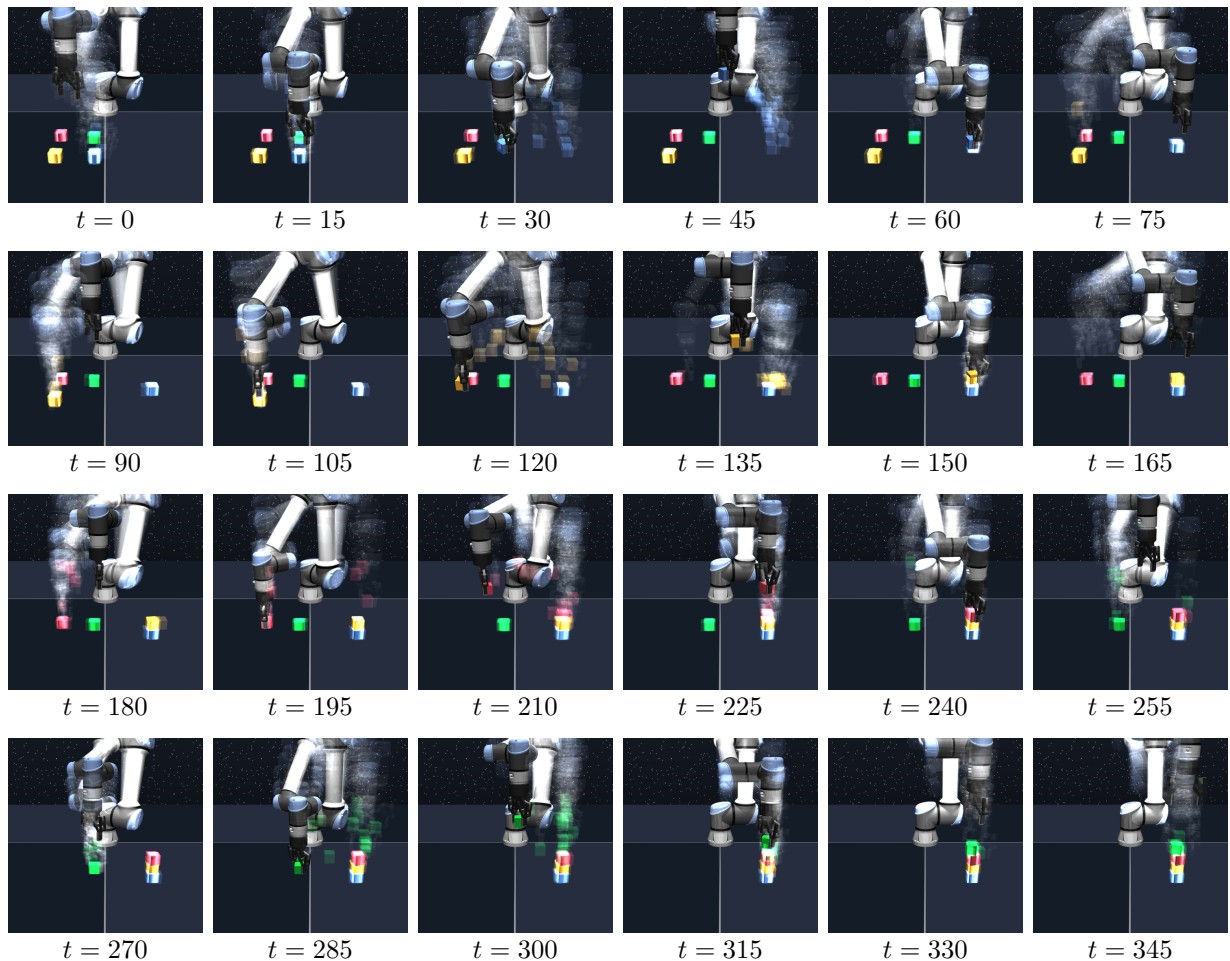

*Figure 11.* Qualitative visualization for an episode of CUBE-4-TASK-5. The robots and cubes shown with transparency indicated GHM samples from the first policy in the GSP.

# F. Experimental Details

## F.1. Base Policies & Hyperparameters

Each policy class is chosen to highlight a different aspect of the pipeline (i.e., GHM learning and skill planning).

1. **Goal-Conditioned TD3 (GC-TD3; Pirotta et al., 2024)**: A standard goal-conditioned offline RL algorithm where we employ Flow Q-learning (FQL; Park et al., 2025c), a flow-based variant of TD3-BC (Fujimoto & Gu, 2021) for policy extraction. Due to bootstrapping on its own learned policy, actions may drift out-of-distribution (OOD) from the dataset, posing a challenge for off-policy predictive modeling (Levine et al., 2020). In particular, we use the standard TD3 critic loss (Fujimoto et al., 2018) to learn a goal-conditioned action-value function $Q_\phi(S, A, G)$ (Schaul et al., 2015):

$$\ell(\phi) = \mathbb{E}_{\substack{(S,A,S',G) \\ A' \sim \pi_G(\cdot|S')}} \left[ \left( Q_\phi(S, A, G) - \mathbb{I}\{S' = G\} - \gamma Q_{\bar{\phi}}(S', A', G) \right)^2 \right] ,$$

where $(S, A, S')$ are transition sampled uniformly from the dataset, and $G$ is a goal state drawn from the following mixture distribution: with probability $0.2$, set $G = S'$, with probability $0.3$ sample $G$ uniformly from the dataset and with $0.5$ set $G$ to a randomly selected future state along the trajectory starting from $(S, A)$, where the selection time step is drawn from a $\gamma$-geometric distribution.

   For policy training, we use the flow Q-learning procedure. We learn jointly a behavior policy $\pi(\cdot \mid S)$ and goal-conditioned policy $\pi_G(\cdot \mid S)$ parametrized by flow-matching vector field. The behavioral policy is trained by a standard conditional flow-matching objective (Lipman et al., 2023) on state-action pairs uniformly sampled from the dataset. The goal-conditioned policy is modeled as a one-step flow map. We denote by $\mu_\psi(S, X_0)$ and $\mu_\omega(S, G, X_0)$ the flow-maps of $\pi$ and $\pi_G$, respectively.

$$\ell(\psi) = -\mathbb{E}_{\substack{(S,A,G) \\ X_0 \sim \mathcal{N}(0,I)}} \left[ Q(S, \mu_\psi(S, G, X_0), G) + \lambda \| \mu_\psi(S, G, X_0) - \mu_\omega(S, X_0) \|^2 \right] , \tag{19}$$

   where $\lambda$ is the distillation coefficient that controls the behavior cloning regularization. Here $G$ is a goal state drawn from the following mixture distribution: with probability $0.5$ sample $G$ uniformly from the dataset, and with $0.5$ set $G$ to a randomly selected future state along the trajectory starting from $(S, A)$.

2. **Goal-Conditioned 1-Step RL (GC-1S)**: A more conservative variant of GC-TD3 that bootstraps using the behavior policy via the dataset's actions. We expect this to yield easier-to-model occupancies as we no longer query the learned value function with OOD actions. Policy extraction on the resulting value function is also performed using FQL (19). In particular, we follow the same training as explained for TD3, and we change only the critic objective:

$$\ell(\phi) = \mathbb{E}_{(S,A,S',A',G)} \left[ \left( Q_\phi(S, A, G) - \mathbb{I}\{S' = G\} - \gamma Q_{\bar{\phi}}(S', A', G) \right)^2 \right] ,$$

   where $A'$ is now sampled from the dataset rather than from the learned policy.

3. **Contrastive RL (CRL; Eysenbach et al., 2022)**: An alternative value-based approach that uses contrastive learning to approximate the successor measure of the behavior policy (Eysenbach et al., 2022). In particular CRL learns a state-action $\phi(s, a)$ and goal encoder $\psi(g)$ by the following Monte-Carlo InfoNCE (van den Oord et al., 2018) contrastive loss:

$$\ell(\phi, \psi) = \mathbb{E}_{(S,A,G)} \left[ -\phi(S, A)^\top \psi(G) \right] - \left[ \log \sum_{(S,A,G')} \exp \left( \phi(S, A)^\top \psi(G') \right) \right] \tag{20}$$

   where $G$ is $\gamma$-distributed sampled future state along the trajectory starting from $(S, A)$ and $G'$ is a state distribued uniformly from the dataset. Moreover, policy extraction is performed using FQL (19) with Q-function $Q(S, A, G)$ estimated as $\phi(S, A)^\top \psi(G)$.

4. **Goal-Conditioned Behavior Cloning (GC-BC; Lynch et al., 2019; Ghosh et al., 2021)**: A purely imitative policy trained via flow matching (Lipman et al., 2023) to mimic the high-quality trajectories in the dataset to reach specific goals using hindsight relabeling (Andrychowicz et al., 2017). Specially, we parametrize the policy by vector field $v_\phi(t, S, G)$

$$\ell(\phi) = \mathbb{E}_{\substack{t,S,A,G \\ X_0 \sim \mathcal{N}(0,I)}} \left[ \| v_\phi(t, S, G) - (A - X_0) \|^2 \right] , \tag{21}$$

where $G$ is $\gamma$-distributed sampled future state along the trajectory starting from $(S, A)$. A key consequence of this value-free approach is that the policy struggles to generalize to distant goals, which can limit the effectiveness of proposing goal-directed "waypoints" during planning.

5. **Hierarchical Flow Behavior Cloning (HFBC; Park et al., 2025b)**: an imitative approach that trains two policies: a high-level policy is trained to predict subgoals that are $h$ steps away from the current state for a fixed lookahead $h$, and the low-level policy is trained to predict actions to reach the given subgoal. Specially we parametrize both high-level and low-level policy by two vector fields $v_\phi(t, S, G)$ and $\nu_\psi(t, S, G)$ respectively.

$$\ell(\phi) = \mathbb{E}_{\substack{t, S_n, S_{n+h}, G \\ X_0 \sim \mathcal{N}(0,I)}} \left[ \|v_\phi(t, S_n, G) - (S_{n+h} - X_0)\|^2 \right],$$

$$\ell(\psi) = \mathbb{E}_{\substack{t, S_n, A_n, S_{n+h} \\ X_0 \sim \mathcal{N}(0,I)}} \left[ \|v_\phi(t, S_n, S_{n+h}) - (A_n - X_0)\|^2 \right],$$

where $S_{n+h}$ is the h-steps way from the current state $S_n$ and $G$ is is $\gamma$-distributed sampled future state along the trajectory starting from $S_n$ Consequently, the high-level policy can be used as a proposal for sub-goal distribution at planning time.

*Table 11.* Base policy hyperparameters. Parameters in { } denote sweeps performed over the values inside brackets.

| Method | Hyperparameter | Antmaze | Cube |
|---|---|---|---|
| **CRL** **GC-TD3** **GC-1S** | FQL Distillation coefficient | $\{0.1, 0.15, 0.2, 0.3, 0.4\}$ | $\{0.7, 0.8, 0.9, 1, 3\}$ |
| | Discount factor | $\{0.995, 0.997\}$ for giant 0.99 otherwise | $\{0.99, 0.995\}$ for CUBE-4 0.99 otherwise |
| | Gradient steps | $\{1M, 3M\}$ | 500k for cube-$\{1, 2\}$ 1M for CUBE-3 3M for CUBE-4 |
| **GC-BC** | Discount factor | $\{0.99, 0.995\}$ for giant $\{0.98, 0.99\}$ for medium $\{0.98, 0.99\}$ for large | $\{0.95, 0.96\}$ for CUBE-1 $\{0.96, 0.97\}$ for CUBE-2 $\{0.96, 0.97, 0.98\}$ for CUBE-3 $\{0.96, 0.98, 0.99\}$ for CUBE-4 |
| | Gradient steps | 125k | 500k for CUBE-1 1M for CUBE-2 2M for CUBE-3 3M for CUBE-4 |
| **HFBC** **SHARSA** | Lookahead | $[25, 50]$ | $[25, 50]$ |
| | Discount factor | $\{0.99, 0.995\}$ | $\{0.95, 0.99, 0.995\}$ |
| | Gradient steps | 3M | 1M for cube-$\{1, 2\}$ 3M for cube-$\{3, 4\}$ |

## F.2. Geometric Horizon Model Hyperparameters

*Table 12.* Hyperparameters for Geometric Horizon Model pre-training.

| | Hyperparameter | Value |
|---|---|---|
| Flow Matching (Lipman et al., 2023) | Probability Path | Conditional OT ($\sigma{=}0$) |
| | Time Sampler | $\mathcal{U}([0,1])$ |
| | ODE Solver | Euler |
| | ODE $\mathrm{d}t$ (train) / steps | 0.1 / 10 |
| | ODE $\mathrm{d}t$ (eval) / steps | 0.05 / 20 |
| Network (U-Net) (Ronneberger et al., 2015) | $t$-Positional Embedding Dim. | 256 |
| | $t$-Positional Embedding MLP | $(1024, 1024)$ |
| | Hidden Activation | mish (Misra, 2019) |
| | Blocks per Stage | 1 |
| | Block Dimensions | $(1024, 1024, 1024)$ |
| Conditional Encoder | Encoder MLP | $(1024, 1024, 1024)$ |
| | Encoder Activation | mish (Misra, 2019) |
| | Conditioning Mixing | additive |
| Optimizer Adam (Kingma & Ba, 2015) | Learning Rate | $10^{-4}$ |
| | Weight Decay | 0 |
| | Gradient Norm Clip | — |
| Training | Max Discount $\gamma_{\mathrm{max}}$ | 0.996 |
| | Target Network EMA | $5 \times 10^{-4}$ |
| | Gradient Steps | 3M |
| | Batch Size | 256 |
| | Context Drop Probability ($z = \varnothing$) | 0.1 |
| | Consistency Proportion | $\begin{cases} 0.25 & \text{for ANTMAZE} \\ 0.15 & \text{for CUBE} \end{cases}$ |
| GCRL Goal Sampling | $p(\text{trajectory goal})$ | 0.5 |
| | $p(\text{random goal})$ | 0.5 |
| | Geometric Trajectory Discount | 0.995 |

## F.3. Planning Hyperparameters

*Table 13.* CompPlan hyperparameters for the main results of the paper.

| Method | Domain | Candidates | Effective horizons | Proposal Distribution | Eval samples | Replan every | Discount |
|--------|--------|-----------|-------------------|----------------------|-------------|-------------|----------|
| CRL | ANTMAZE-MEDIUM | 256 | $[50, 50, 100, 100, 200]$ | Conditional | 256 | 1 | 0.999 |
| | ANTMAZE-LARGE | 256 | $[50, 50, 100, 100, 200]$ | Conditional | 256 | 1 | 0.999 |
| | ANTMAZE-GIANT | 256 | $[50, 50, 100, 100, 200]$ | Conditional | 256 | 1 | 0.999 |
| | CUBE-1 | 1024 | $[20, 80]$ | Unconditional | 128 | 1 | 0.99 |
| | CUBE-2 | 1024 | $[20, 20, 80]$ | Unconditional | 128 | 1 | 0.99 |
| | CUBE-3 | 1024 | $[20, 20, 20, 80]$ | Unconditional | 128 | 1 | 0.99 |
| | CUBE-4 | 1024 | $[20, 20, 20, 20, 80]$ | Unconditional | 128 | 1 | 0.99 |
| GC-TD3 | ANTMAZE-MEDIUM | 256 | $[50, 50, 100, 100, 200]$ | Conditional | 256 | 1 | 0.999 |
| | ANTMAZE-LARGE | 256 | $[50, 50, 100, 100, 200]$ | Conditional | 256 | 1 | 0.999 |
| | ANTMAZE-GIANT | 256 | $[50, 50, 100, 100, 200]$ | Conditional | 256 | 1 | 0.999 |
| | CUBE-1 | 1024 | $[20, 80]$ | Unconditional | 128 | 1 | 0.99 |
| | CUBE-2 | 1024 | $[20, 20, 80]$ | Unconditional | 128 | 1 | 0.99 |
| | CUBE-3 | 1024 | $[20, 20, 20, 80]$ | Unconditional | 128 | 1 | 0.99 |
| | CUBE-4 | 1024 | $[20, 20, 20, 20, 80]$ | Unconditional | 128 | 1 | 0.99 |
| GC-BC | ANTMAZE-MEDIUM | 256 | $[50, 50, 100, 100, 200]$ | Conditional | 256 | 1 | 0.999 |
| | ANTMAZE-LARGE | 256 | $[50, 50, 100, 100, 200]$ | Conditional | 256 | 1 | 0.999 |
| | ANTMAZE-GIANT | 256 | $[50, 50, 100, 100, 200]$ | Conditional | 256 | 1 | 0.999 |
| | CUBE-1 | 1024 | $[20, 80]$ | Unconditional | 128 | 1 | 0.99 |
| | CUBE-2 | 1024 | $[20, 20, 80]$ | Unconditional | 128 | 1 | 0.99 |
| | CUBE-3 | 1024 | $[20, 20, 20, 80]$ | Unconditional | 128 | 1 | 0.99 |
| | CUBE-4 | 1024 | $[20, 20, 20, 20, 80]$ | Unconditional | 128 | 1 | 0.99 |
| GC-1S | ANTMAZE-MEDIUM | 256 | $[50, 50, 100, 100, 200]$ | Conditional | 256 | 1 | 0.999 |
| | ANTMAZE-LARGE | 256 | $[50, 50, 100, 100, 200]$ | Conditional | 256 | 1 | 0.999 |
| | ANTMAZE-GIANT | 256 | $[50, 50, 100, 100, 200]$ | Conditional | 256 | 1 | 0.999 |
| | CUBE-1 | 1024 | $[20, 80]$ | Unconditional | 128 | 1 | 0.99 |
| | CUBE-2 | 1024 | $[20, 20, 80]$ | Unconditional | 128 | 1 | 0.99 |
| | CUBE-3 | 1024 | $[20, 20, 20, 80]$ | Unconditional | 128 | 1 | 0.99 |
| | CUBE-4 | 1024 | $[20, 20, 20, 20, 80]$ | Unconditional | 128 | 1 | 0.99 |
| HFBC | ANTMAZE-MEDIUM | 32 | $[25] * 24$ | Conditional | 128 | 1 | 0.999 |
| | ANTMAZE-LARGE | 32 | $[25] * 24$ | Conditional | 128 | 1 | 0.999 |
| | ANTMAZE-GIANT | 32 | $[25] * 24$ | Conditional | 128 | 1 | 0.999 |
| | CUBE-1 | 32 | $[25] * 4$ | Unconditional | 128 | 1 | 0.99 |
| | CUBE-2 | 32 | $[25] * 5$ | Unconditional | 128 | 1 | 0.99 |
| | CUBE-3 | 32 | $[25] * 6$ | Unconditional | 128 | 1 | 0.99 |
| | CUBE-4 | 32 | $[25] * 7$ | Unconditional | 128 | 1 | 0.99 |

