# OpenReview forum: "Compositional Planning with Jumpy World Models"
_ICML.cc/2026/Conference — ICML 2026 regular_

### Official Review · Reviewer_FLJ3 · 2026-03-12

**Soundness:** 3
**Presentation:** 3
**Significance:** 3
**Originality:** 3
**Overall Recommendation:** 5
**Confidence:** 4

**Summary:**

This paper proposes a framework for compositional planning using jumpy world models that enables agents to sequence pre-trained policies as temporally extended actions. Authors argue that foundation policies, while capable in many settings, struggle with long-horizon tasks that require combining behaviors. To address this limitation, the paper introduces a (policy, horizon)-conditioned jumpy world model capable of predicting successor state distributions across multiple time scales.

The method builds upon the Temporal Difference Flow framework and introduces a horizon consistency objective (TD-HC) designed to align predictions across discount factors and improve long-horizon predictive accuracy. Using this model, the authors derive an estimator that evaluates the value of executing arbitrary sequences of policies with variable durations, formalized through geometric switching policies.

The resulting planning algorithm, COMPPLAN, searches for sequences of policies that maximize expected return. The method is evaluated on long-horizon robotic manipulation and navigation tasks from the OGBench benchmark using several classes of base policies. Results demonstrate large improvements over zero-shot policies, action-level planning, and hierarchical RL baselines.

**Compliance With Llm Reviewing Policy:**

Affirmed.

**Final Justification:**

After the rebuttal, my main questions were addressed clearly, especially regarding the distinction from Dreamer/TD-MPC-style world models, the relevance of skill-based methods in the OGBench setting, the robustness of planning to moderate model error, and the applicability of the framework to broader policy classes such as language-conditioned policies.

I also found the additional humanoidmaze results helpful. While they do not fully close the gap to broader embodied or multimodal benchmarks, they strengthen the evidence that the benefits of compositional planning persist in larger and longer-horizon settings.

Overall, the rebuttal increased my confidence in both the positioning and the empirical relevance of the work. I continue to view this as a strong paper with meaningful technical and practical contributions, while noting that evaluation beyond the current benchmark family would further strengthen the claims.

**Key Questions For Authors:**

- Can the proposed framework scale to larger settings, such as composing pre-trained policies from Vision-Language-Action (VLA) models or other foundation robotics policies?
- Could COMPPLAN be applied to robotics manipulation benchmarks such as LIBERO, where offline datasets include text-conditioned tasks and evaluation can be performed against foundation policies? It would be interesting to see whether the proposed compositional
planning approach can leverage language-conditioned policies in such environments.
- During inference, does the planning procedure rely on reward signals obtained from the simulator when evaluating candidate policy sequences? If so, how robust is the method when state predictions from GHM are inaccurate, potentially leading to incorrect reward estimates during planning?

**Limitations:**

- While COMPPLAN is demonstrated on goal-conditioned RL policies in OGBench, the evaluation is limited to state- or goal-conditioned settings. The work could be stronger if the method were applied to Vision-Language-Action (VLA) policies, where the goal is specified through text instructions rather than explicit state goals.

**Strengths And Weaknesses:**

Strengths
- The proposed framework demonstrates strong performance on long-horizon tasks where foundation policies often fail. The extension of Temporal Difference Flows with the TD-HC horizon consistency objective appears mathematically well motivated and provides a principled approach to improving long-horizon predictive accuracy.
- The paper introduces a framework for direct compositional planning over parameterized policies without requiring task-specific training. This represents a meaningful departure from traditional hierarchical reinforcement learning methods, which typically rely on learning task-dependent hierarchies.
- The experimental results support the proposed approach and show strong performance improvements. The evaluation is reasonably comprehensive, covering multiple policy families and domains (ANTMAZE and CUBE tasks), and includes comparisons with relevant baselines such as action-level planning, generalized policy improvement, and hierarchical reinforcement learning methods.

Weaknesses
- The manuscript could better clarify how the proposed method differs from existing world-model approaches. In particular, it would be useful to explain why one should use geometric horizon models instead of alternative world models such as those used in Dreamer[1] or reconstruction-free world models like TD-MPC [2].
- While Section 4.3 compares multi-timescale GHMs to a one-step world model trained with flow matching, it would be valuable to include comparisons with established world-model methods (e.g., Dreamer-style models) to further contextualize the benefits of the proposed approach.
- The related work discussion in the main manuscript is limited. In particular, comparisons with skill-based model-based RL methods (such as SkiMo[3]) appear important for clearly distinguishing the proposed framework from existing approaches.
- Although COMPPLAN is evaluated with several goal-conditioned base policies, comparisons with skill-based model-based RL methods (in addition to HIQL and SHARSA) would further strengthen the empirical evaluation.
- Broader domain diversity would strengthen the results. Evaluating the method on additional robotic manipulation benchmarks such as LIBERO or MetaWorld could help demonstrate generality.
- The paper would benefit from a clearer discussion of the computational cost of training GHMs.

[1] Mastering Diverse Domains through World Models. Nature 2025

[2] TD-MPC2: Scalable, Robust World Models for Continuous Control. ICLR 2024

[3] Skill-based Model-based Reinforcement Learning. CoRL 2022

---

> ### Author Rebuttal · Authors · 2026-03-31
>
> We sincerely thank the reviewer for the thoughtful evaluation and recognition of our contributions. We hope to address your concerns below.
>
> ---
>
> > Comparison to Dreamer / TD-MPC
>
> Our method addresses a fundamentally different setting. Dreamer and TD-MPC learn one-step dynamics and plan over primitive actions, training both a world model and policy from scratch for a specific task. Our setting is different: we assume access to a repertoire of pre-trained policies and ask how to compose them to solve downstream tasks, closer to a form of hierarchical planning over temporally extended actions (sometimes called macro-actions).
>
> This difference dictates what the model must predict. Planning over policies requires predicting the distribution of states a policy will visit, not just the effect of one primitive action. GHMs model exactly this: the outcome of rolling out a policy under a geometrically distributed horizon. This provides a principled task-agnostic asbraction of a policies behavior that can be reused and composed to solve downstream tasks without retraining.
>
> ---
>
> > Skill-based MBRL (SkiMo, etc.) and empirical comparisons
>
> OGBench is an offline goal-conditioned RL benchmark, which imposes unique constraints that render many of the suggested methods inapplicable. SkiMo requires online interaction after its pre-training procedure. Dreamer and TD-MPC are task-dependent and require online interaction with the environment. Diffusion-based planners have only been demonstrated in navigation tasks over restrictive state representations (e.g., [x,y] coordinates) and haven’t scaled to manipulation. Within the methods applicable to our setting, we compare against HIQL and SHARSA – the current state-of-the-art on OGBench – and outperform both substantially, particularly on cube-4 (67% vs. 9% for SHARSA, 0% for HIQL).
>
> ---
>
> > Domain diversity and language-conditioned policies
>
> OGBench was chosen because it systematically scales task complexity through both horizon length and combinatorial difficulty, making it well-suited for isolating the benefits of compositional planning. Benchmarks like LIBERO and MetaWorld introduce vision and language modalities, which would require integrating representation learning / learned representations into our pipeline. While these are solvable challenges, we view this as an orthogonal and exciting direction for future work. We deliberately scoped the current work to validate that compositional planning with GHMs can handle complex, long-horizon tasks before layering on additional modeling decisions around representation learning. That said, nothing in our framework is specific to OGBench or state-based observations; CompPlan applies wherever parameterized policies are available.
>
> ---
>
> > Scalability to VLA / foundation policies
>
> CompPlan is policy-class agnostic. For VLAs, text instructions naturally serve as the conditioning variable $z$: a GHM conditioned on z="open the drawer" would learn to predict the distribution of future states induced by executing that language-conditioned policy (i.e., the VLA). To summarize, CompPlan requires three ingredients: (i) a collection of parameterized policies $\pi_z$ from which we can sample actions, (ii) an offline dataset of transitions (which need not come from the policies themselves, as GHMs learn off-policy), and (iii) a reward function or scoring mechanism (e.g., distance function, success classifier, VLM-based reward) to evaluate candidate policy sequences at planning time.
>
> ---
>
> > Sensitivity of planning to reward accuracy
>
> Planning uses $r$ to score candidate sequences via Lemma 1 – requiring only $r(s)$ at GHM-sampled states, not simulator access. Because CompPlan queries GHMs only on in-distribution controls (pre-existing policies), prediction error empirically remains under control. Table 3 and 9 confirm: TD-HC reduces EMD by up to 28% yet planning success hardly changes, showing the robustness to moderate inaccuracies.

---

> > ### Author Rebuttal · Reviewer_FLJ3 · 2026-04-01
> >
> > I acknowledge that direct comparisons with model-based RL methods such as Dreamer or skill-based methods may be challenging. Most of my concerns are addressed, and I will keep my positive rating.

---

> > > ### Author Response · Authors · 2026-04-05
> > >
> > > We sincerely thank the reviewer for the engaged and constructive discussion. We are glad to hear that our responses have adequately addressed your concerns.
> > >
> > > To briefly recap the contributions: CompPlan introduces a principled, training-free framework for composing pre-trained policies via jumpy world models, demonstrating strong improvements across a range of challenging long-horizon tasks. The TD-HC consistency objective is theoretically grounded and empirically validated, and the framework unifies several existing paradigms (GPI, action-level planning, GGPI) under a single formulation. Additionally, motivated in part by your suggestion for broader domain diversity, during the rebuttal period we ran new experiments on the humanoidmaze domain, which we think you'd appreciate --- these feature higher dimensional state-spaces and longer horizons:
> > >
> > >
> > >
> > > |                             |   HFBC         |   CompPlan |
> > > |:----------------------------|--------------:|------------:|
> > > | humanoidmaze-medium        |          0.80 |         **0.96** |
> > > | humanoidmaze-large         |          0.40 |         **0.64** |
> > > | humanoidmaze-giant      |  0.34    | **0.68** |
> > >
> > > Consistent with our other results, the gains from compositional planning grow as the horizon increases -- doubling success rates on the giant maze. We believe this further demonstrates the generality of our approach.
> > >
> > > Given the progress made during this discussion, we hope the reviewer will consider whether the final score reflects the current state of the paper. Either way, we are grateful for your time and thoughtful feedback.

---

### Official Review · Reviewer_th6z · 2026-03-12

**Soundness:** 3
**Presentation:** 3
**Significance:** 3
**Originality:** 3
**Overall Recommendation:** 5
**Confidence:** 4

**Summary:**

The manuscript introduces a technically sophisticated framework for compositional planning using multi-timescale predictive models of policy behavior. The proposed approach integrates successor-measure modeling, temporal abstraction, and generative world models, which presents a coherent theoretical formulation and empirical validation. Experimental results show strong improvements on long-horizon tasks relative to several baselines.

**Compliance With Llm Reviewing Policy:**

Affirmed.

**Final Justification:**

The authors' rebuttal has addressed my questions and concerns. I appreciate it. Therefore, I increased my score from 4 to 5.

**Key Questions For Authors:**

Please see the weakness section.

**Limitations:**

There is only one sentence on limitations, and no section on potential negative societal impact. Please address those.

**Strengths And Weaknesses:**

### Strengths

- The paper addresses the important challenge of planning with temporal abstractions by composing pre-trained policies rather than primitive actions
- The method combines successor measures, geometric switching policies, and flow-based generative modeling into a unified framework. The decomposition of successor measures for composite policies provides a principled basis for estimating the value of policy sequences.
- The proposed TD-HC loss enforces coherence between predictions at different discount factors, addressing instability in long-horizon predictions. This contribution is technically meaningful and grounded in a derived Bellman-like relationship across horizons.
- Results on OGBench demonstrate consistent improvements over zero-shot policies, action-level planning, and hierarchical baselines, with especially large gains on complex tasks such as ANTMAZE-GIANT and CUBE-4.

### Weakness
- While the paper proposes an estimator for evaluating policy sequences, there is limited analysis of how model prediction errors affect planning quality. Can the authors comment on whether it's possible to establish formal guarantees or bounds on performance degradation under inaccurate successor measure predictions?
- The approach assumes access to a diverse library of pretrained policies. The manuscript does not fully discuss the sensitivity of planning performance to the diversity or quality of these policies.
- The experimental comparisons include hierarchical RL baselines and one-step world models but do not extensively compare with recent diffusion-based planners, latent world models, or model-based RL systems that also address long-horizon prediction.

---

> ### Author Rebuttal · Authors · 2026-03-31
>
> We thank the reviewer for their positive assessment and recognition of our contribution. We hope to address each of your concerns below.
>
> ---
>
> > Limited analysis of how model prediction errors affect planning quality.
>
> We agree that formal guarantees connecting model accuracy to planning performance would be a valuable contribution. However, establishing such bounds in a convincing and practical setting is challenging because planning performance depends on many interacting factors beyond model predictive accuracy alone – including the quality of the proposal distribution, the choice of effective horizons, the number of composition steps, and whether the model is queried in- or out-of-distribution. Recent work [1] has empirically confirmed this view in the broader model-based RL setting, showing that accurate prediction alone is insufficient to explain the success of planning.
>
> Empirically, our results illustrate this nuance directly. Table 3 shows that TD-HC reduces EMD by up to 28% compared to TD-Flow at long horizons, yet Table 9 reveals that planning success rates are nearly identical, indicating that the planning procedure is robust to moderate model inaccuracies. We will expand the discussion to elaborate on the many factors influencing planning quality and identify formal analysis as an important direction for future work.
>
> ---
>
> > Sensitivity of planning performance to the diversity or quality of base policies.
>
> We believe our paper directly addresses this question through its experimental design. We intentionally evaluate five policy classes spanning value-based, contrastive, imitative, and hierarchical approaches – each with different strengths, weaknesses, and inductive biases. Section 4.2 analyzes this extensively, showing that CompPlan improves over all policy types and that weak zero-shot policies (e.g., GC-BC at 0% on cube-4) can become highly effective when composed (76% with planning).
>
> Additionally, the appendix includes several ablations over planning hyperparameters. For example, the ablation on planning objectives shows that — when planning with CRL policies in cube — jointly optimizing the initial actions can substantially outperform optimizing only the policy encoding $z$. This is because CRL policies are relatively diffuse and highly stochastic: they learn a representation space that does not localize cube positions well, which makes planning in $z$-space alone less effective.
>
> We will revise the paper to make this form of sensitivity analysis more prominent.
>
> ---
>
> > Comparisons with diffusion planners / model-based RL methods
>
> Thank you for this suggestion. We discuss these methods in Appendix A, but we appreciate the opportunity to elaborate on the key distinctions here. We believe the gap between our approach and diffusion-based trajectory planners is fundamental. Diffusion planners generate trajectory segments without any understanding of what a policy can actually accomplish in the environment, they potentially propose waypoints that no policy can reliably reach. Our framework is built around affordances: because we model the successor measure of each policy, we understand precisely how every policy evolves in the environment, and only ever plan over realizable behaviours. To compensate for this lack of affordance awareness, diffusion planners rely on simplifying assumptions such as oracle goal/observation representations. Notably, these methods have only been demonstrated on navigation tasks with these restrictive assumptions and haven’t been shown to scale to complex manipulation domains (e.g., MCTS diffusion [2; Table 2] obtains 0% success in cube-4 even under these simplifying assumptions).
>
> More broadly, our work addresses a distinct problem – composing pre-trained policies as temporally extended building blocks – which is something these trajectory-level methods are not designed to do. We will revise the main text to better surface these distinctions.
>
> Regarding model-based RL, we refer to the first part of the rebuttal for Reviewer FLJ3 which further expands on this. That said, the ActionPlan baseline is fundamentally a world model planning comparison but with action proposals coming from the behavior policy to avoid out-of-distribution queries. CompPlan achieves a 200% relative improvement over ActionPlan on long-horizon tasks, demonstrating the clear advantage of planning over temporally extended behaviors rather than primitive actions.
>
> ---
>
> > Limitations and societal impact.
>
> We will revise the manuscript to include a more thorough discussion of the limitations.
>
> ---
>
> ### References
>
> [1] The Surprising Difficulty of Search in Model-Based Reinforcement Learning. Wei-Di Chang, Mikael Henaff, Brandon Amos, Gregory Dudek, Scott Fujimoto. CoRR abs/2601.21306. 2026.
>
> [2] Monte Carlo Tree Diffusion for System 2 Planning. Jaesik Yoon, Hyeonseo Cho, Doojin Baek, Yoshua Bengio, Sungjin Ahn. CoRR abs/2502.07202. 2025.

---

> > ### Author Rebuttal · Reviewer_th6z · 2026-04-03
> >
> > Thank you for the detailed responses. I believe that this is a very interesting and novel contribution. While your clarifications on empirical robustness to model inaccuracies are helpful, some key concerns regarding theoretical guarantees and broader empirical comparisons remain, so I maintain my weak accept recommendation.

---

> > > ### Author Response · Authors · 2026-04-05
> > >
> > > We thank the reviewer for the positive assessment and for considering our work to be "very interesting and novel." We appreciate the candid feedback on the remaining concerns and wanted to briefly address both.
> > >
> > > Regarding theoretical guarantees on planning under model inaccuracies -- we share the reviewer's interest in this direction. As noted in our rebuttal, recent work [1] has shown that even in standard model-based RL, prediction accuracy alone does not explain planning success, suggesting that such guarantees remain an open challenge for the field broadly. We view this as an important avenue for future work rather than a limitation specific to our framework.
> > >
> > > On broader empirical comparisons, we wanted to highlight that the most relevant diffusion-based planner (MCTS Diffusion [2]) obtains 0% on cube-4 even under simplifying assumptions, while CompPlan achieves 67% without such restrictions. Additionally, since the initial submission, we have extended our evaluation to the humanoidmaze domains -- which feature significantly higher-dimensional state spaces and longer horizons:
> > >
> > >
> > > |                             |   HFBC         |   CompPlan |
> > > |:----------------------------|--------------:|------------:|
> > > | humanoidmaze-medium        |          0.80 |         **0.96**|
> > > | humanoidmaze-large         |          0.40 |         **0.64**|
> > > | humanoidmaze-giant      |  0.34    | **0.68** |
> > >
> > > CompPlan's gains grow with horizon length, doubling the success rate on the giant maze. We hope this helps address the "limited evaluation" concern noted in the recommendation.
> > >
> > > Thank you again for the thoughtful engagement throughout this process.

---

### Official Review · Reviewer_f7Yr · 2026-03-13

**Soundness:** 4
**Presentation:** 4
**Significance:** 4
**Originality:** 4
**Overall Recommendation:** 6
**Confidence:** 4

**Summary:**

This paper proposes a planning framework called CompPlan that treats a collection of pre-trained policies  indexed by some conditioning variable $z$ as temporally extended actions. It seeks to optimize the value of a trajectory consisting of taking action $a_1$ then following a sequence of $N$ policies $\pi_{z_1},\ldots, \pi_{z_n}$ with corresponding switching probabilities $\{\alpha_1, \ldots, \alpha_{n-1}\}$ that control how long each policy is executed before switching to the next. Importantly, the switching probability induces a geometric distribution over the length of execution, which can be combined with the global discount factor to produce an “effective” total discount.

To plan across a sequence of policies, they train a generative model of the successor measure (GHM), conditioned both on a policy identifier $z$ and a discount factor $\gamma$, which allows the planner to generate successor states for any policy $z$ at any global discount $\gamma$ and switching probability $\alpha$. Intuitively, a high switching probability means that the policy is very likely to terminate early, and corresponds to a lower effective discount factor that generates more temporally close states.

To train this measure, they generalize the previous TD-Flow method with an objective that encourages Bellman-like consistency across different discount factors, rather than learning each horizon independently.

Finally, to plan with chained policies, we need to be able to evaluate candidate sequences. To estimate the value of such a chain of policies with variable execution times, the authors show that chaining *samples* from each policy’s successor measure and summing the rewards at these states, weighted by the probability that the agent would still be executing that policy at the $k$th policy stage, yields an unbiased estimator of the state-action value of the full geometric switching policy, giving a planning criterion.

In practice, the paper uses policies conditioned on subgoals $z \in S$ proposed by the learned GHM. To generate candidate plans given a goal $g$, they sample a sequence of plans from a proposal distribution and produce rollouts by repeatedly querying the GHM for successor states with the previously predicted state $z_{t-1}$ and actions produced by the policy $\pi_g(z_{t-1})$ conditioned on $z_{t-1}$ and $g$.

**Compliance With Llm Reviewing Policy:**

Affirmed.

**Final Justification:**

My primary concerns were about (1) missing evaluations on the most long-horizon environments where this approach should shine, and (2) room for improvement by optimizing the proposal distribution. (1) was addressed with humanoid-giant experiments and I thought the answer to (2) was satisfactory. The ablations were very thorough and there it took many well-justified components to create  a cohesive paper, so I am raising my score to a 6.

**Key Questions For Authors:**

1. Why did the authors choose not to evaluate on the ``humanoidmaze`` environment?
2. Is it straightforward to optimize the proposal distribution over the sequence of policies, either at test time or during training? What additional difficulties might this form of (more flexible) planning introduce?
3. Is training a discount-conditioned GHM significantly more expensive that one with a fixed discount factor? How much more so?

**Limitations:**

Yes

**Strengths And Weaknesses:**

**Strengths**

This work proposes several novel ideas, including variable switching probabilities for geometric policy composition, the notion of effective discount factors to consider multiple timescales of execution, and an unbiased estimator of the value of such a compositional policy, and also introduce practical methods to efficiently train discount-conditioned GHMs. Overall, this is a strong work that paves the way for practical instantiation for combining options with variable termination conditions.

The empirical results, especially on compositional tasks like cube-2 to cube-4, are impressive. and the ablations are comprehensive and demonstrate the contribution of each component (e.g., the consistency objective, planning horizon, etc.).

**Weaknesses**

Given the paper's focus on composing pretrained policies for long-horizon tasks, I would have expected evaluations on the most long-horizon ``humanoidmaze`` environments in OGBench.

---

> ### Author Rebuttal · Authors · 2026-03-31
>
> We sincerely thank the reviewer for the positive and insightful review, and for recognizing the novelty and significance of our contributions.
>
> ---
>
> > Why did the authors choose not to evaluate on the humanoidmaze environment?
>
> Our initial evaluation focused on antmaze and cube tasks as they provide a diverse testbed spanning navigation and manipulation with varying horizon lengths. That said, following the reviewer’s suggestion, we have extended our experiments to humanoidmaze and can now report results on humanoidmaze-medium and humanoidmaze-large:
>
> |                             |   HFBC 		|   CompPlan |
> |:----------------------------|--------------:|------------:|
> | humanoidmaze-medium        |          0.80 |         **0.96**|
> | humanoidmaze-large         |          0.40 |         **0.64**|
>
>
> These results are consistent with our findings: CompPlan provides consistent improvements over strong zero-shot baselines. Results for humanoidmaze-giant are still running given the tight rebuttal timeline and will be posted once available.
>
> ---
>
> > Is it straightforward to optimize the proposal distribution over the sequence of policies, either at test time or during training? What additional difficulties might this form of (more flexible) planning introduce?
>
> This is an excellent question. In our current framework, we use random shooting with simple proposal distributions (goal-conditioned or unconditional GHMs), which works well in practice as demonstrated by our ablations in Appendix D.5. However, optimizing the proposal distribution more directly (e.g., MPC methods like CEM, MPPI) is a natural extension that could improve sample efficiency of the planning procedure. The primary difficulty is that, for goal-conditioned policies, the search space corresponds to the full state space, which is high-dimensional and lacks a compact structure amenable to these sampling-based optimizers. A natural solution would be to plan in a learned latent space instead, for instance by leveraging policy classes that naturally operate over compact latent representations such as FB [1] or HILP [2]. In such settings, CEM/MPPI-style refinement over latent sequences becomes tractable. We view this as a promising direction for future work and will expand the discussion accordingly.
>
> ---
>
> > Is training a discount-conditioned GHM significantly more expensive than one with a fixed discount factor? How much more so?
>
> Training a discount-conditioned GHM introduces no meaningful computational overhead compared to a fixed-discount GHM. In practice, the discount factor is just an additional conditioning input, implemented in the same way as the time encoding, whose embedding is simply added to the policy and time embeddings. TD-HC further improves the ability of the GHM to learn across many discount factors by leveraging the Bellman-like relationship between successor measures at different timescales (§3.2), enabling short-horizons to directly supervise long-horizon learning. The overhead of TD-HC is dominated by drawing an additional sample from the GHM, but this cost is low as it is only applied to a small fraction of each mini-batch.
>
> ---
>
> ### References
>
> [1] ​​Ahmed Touati and Yann Ollivier. Learning One Representation to Optimize All Rewards. Neural Information Processing Systems (NeurIPS), 2021.
>
> [2] Seohong Park, Tobias Kreiman, and Sergey Levine. Foundation Policies with Hilbert Representations. International Conference on Machine Learning (ICML), 2024.

---

> > ### Author Rebuttal · Reviewer_f7Yr · 2026-04-04
> >
> > I appreciate the additional experiments on humanoidmaze! While the composition of policies with different switching probabilities induces a very large search space over optimal policies, I believe this is generally an unavoidable problem and find the idea of generalizing FB with some sort of compositional representation of z very interesting. I look forward to seeing the results on the giant maze and am open to raising my score to a strong accept (6).

---

> > > ### Author Response · Authors · 2026-04-05
> > >
> > > We sincerely thank the reviewer for the continued enthusiasm and constructive engagement. We’re glad the humanoidmaze experiments were helpful, and we’re excited to share the humanoidmaze-giant results:
> > >
> > > |                             |   HFBC         |   CompPlan |
> > > |:----------------------------|--------------:|------------:|
> > > | humanoidmaze-medium        |          0.80 |         **0.96**|
> > > | humanoidmaze-large         |          0.40 |         **0.64**|
> > > | humanoidmaze-giant      |  0.34    | **0.68** |
> > >
> > > Consistent with our other results, CompPlan’s gains grow with task horizon, doubling the success rate on the giant maze. This is precisely the regime where compositional planning should shine, and we’re pleased to see it hold up in this higher-dimensional humanoid setting.
> > >
> > > We also appreciate your note on the connection to compositional representations over FB-style latent spaces -- we agree this is a very promising direction to explore in future work.
> > >
> > > Thank you again for the thoughtful review and engaging discussion. We hope you find the evaluation now fully addresses your earlier concerns.

---

### Official Review · Reviewer_QdoD · 2026-03-13

**Soundness:** 3
**Presentation:** 4
**Significance:** 3
**Originality:** 2
**Overall Recommendation:** 5
**Confidence:** 3

**Summary:**

This work introduces CompPlan, a framework for high-level planning over pre-trained policies using jumpy world models. The jumpy world models do not learn next-step prediction but are trained as generative models of successor measures on different jumps or horizons. The method can be decomposed into two parts:  a novel TD-Flow model with horizon-consistency (for the same timestep, the model's predictions with different jumps should be consistent), and a planning formulation for policy sequences with variable stage-wise switching schedules.  The results are structured properly, starting with showing that planning with the learnt model is better than zero-shotting with a pre-trained policies. Then they use their model to evaluate different planning strategies: selecting a single policy, using a single policy, and planning which policy to pick at which horizon, and show that the latter is the best. Finally, they show that their test-time compositional method improves over methods that are optimized for high-level goal selection during training.

**Compliance With Llm Reviewing Policy:**

Affirmed.

**Final Justification:**

The rebuttal fully addressed my concerns about whether Jumpy multi-horizon modeling ⟹ better planning, which is the central claim of the work, and improved my overall assessment. My remaining concern is about Table 9, namely, whether TD-HC is beneficial as the core model compared to a model that is equally good at the short horizon predictions. Clarifying this link could have provided more insight into the role of long-horizon predictability in jumpy or long-horizon planning (apparently, it is still not clear whether it is needed, but the authors did not claim that either). The authors’ explanation is plausible: TD-HC could be better for planning at long horizons, while for shorter ones it offers orthogonal value by improving generative fidelity at the longest horizons. I would regard this as a minor weakness rather than a major issue.

**Key Questions For Authors:**

1. Why is GGPI omitted from the main comparisons, given that it is the closest prior method and already uses GHMs for geometric policy composition?

2. Can the authors provide a short-horizon/one-step world evaluation for td-flow vs td-hc? This would clarify whether the proposed consistency loss addresses a genuine blind spot of short-horizon evaluation or mainly improves very long-horizon occupancy modeling.

3. Is planning with one-step prediction of a GHM beneficial (trained to predict more than one-step but maybe better on one-step prediction)? The use of a separately trained 1-step world model for evaluation is understandable, but it does not isolate the accuracy of a one-step prediction world models from the results in Figure 1.

I can improve my score with more clarity on the novelty and analysis of the planning performance, given the world model accuracy.

**Limitations:**

Yes

**Strengths And Weaknesses:**

### Strengths

1. The paper is well structured, and the need for compositional planning with GHMs is supported by strong ablations over the important components of their model.  The world model performance vs TD-Flow baseline is isolated in Table 3, showing clear improvement across two policies and different tasks. They show that planning with their model is better in general vs pretrained-policies, and specifically against different planning horizon strategies.
2. The presentation and delivery are quite clear to understand, aside from a few parts discussed in weakness.

### Weaknesses
1. The use of a jumpy world model is not itself new, as stated in the paper, as GGPI already builds on Geometric Horizon Models (GHMs) . Therefore, the novelty of the current paper is not the jumpy world model per se, but rather its extension to stage-wise switching schedules, more general policy-sequence evaluation, and the new TD-Flow-based consistency loss. However, the planning strategies ablations do not include GGPI as a strategy, but rather a heuristic over the horizons.

2. The downstream planning of the proposed extra loss is of weak value. The paper shows that the horizon-consistency loss improves long-horizon generative fidelity, but it also reports that the effect on planning performance is limited in the tested regime because planning mainly queries moderate horizons. So one of the main methodological additions appears only weakly beneficial for downstream decision making on the actual benchmarks.

In conclusion, the analysis, in general, is good, but the novelty analysis is not clear.

---

> ### Author Rebuttal · Authors · 2026-03-31
>
> We thank the reviewer for their constructive feedback and appreciation of the paper’s structure and ablations. We hope to address each of your concerns below.
>
> ---
>
> > Why is GGPI omitted from the main comparisons?
>
> GGPI is not so much a competing baseline as it is the starting point we generalize. Our framework imbues geometric switching policies with variable switching probabilities (Theorem 1, Lemma 1) and provides a scalable instantiation of GGPI that operates in a fundamentally different regime. GGPI was demonstrated with four discrete policies, two separately trained VAE-based GHMs at horizons of 5 and 10 steps, and compositions restricted to pairs of policies via exhaustive search. By contrast, we train a single GHM across continuous policy families and a continuum of discount factors with horizons up to 25x longer, and compose sequences of up to 24 policies. Bridging this gap required new conceptual insights, a more powerful generative model, and a tractable planning procedure for continuous policy spaces.
>
> On the last point, GGPI implements policy improvement via exhaustive enumeration – a strategy that is infeasible when the policy set is continuous. Notably, our solution of leveraging the GHMs themselves as the proposal distribution elegantly repurposes the same model used for prediction and introduces no additional learned components.
>
> We hope this clarifies the nature of our contributions relative to GGPI, and we will revise the manuscript to make these distinctions more prominent.
>
> ---
>
> > Can the authors provide a short-horizon evaluation for TD-Flow vs. TD-HC?
>
> We ran additional evaluations at shorter horizons ($\gamma \in \{ 0.98, 0.995 \}$). We report EMD in the following table:
>
> ||TD-Flow|TD-HC|
> |:-|-:|-:|
> |antmaze-medium, crl, 0.98|3.91346|**3.86575**|
> |antmaze-large, crl, 0.98|4.07392|**4.03633**|
> |antmaze-giant, crl, 0.98|4.45069|**4.19618**|
> |antmaze-medium, crl, 0.995|4.40853|**4.21678**|
> |antmaze-large, crl, 0.995|5.23652|**4.80752**|
> |antmaze-giant, crl, 0.995|6.76894|**5.73581**|
>
> As can be seen, TD-HC consistently outperforms TD-Flow across all tested horizons, with the margin shrinking at shorter horizons and growing substantially at longer ones, confirming that the consistency loss provides cumulative benefits as the horizon increases. We also refer the reviewer to Figure 8 in the appendix for a qualitative illustration: at $\gamma=0.99$ the two methods produce visually similar occupancy distributions, but at $\gamma=0.998$ TD-Flow generates samples that erroneously traverse walls and fails to capture the tail of the distribution, while TD-HC maintains physically plausible distributions and captures the correct tail.
>
> We foresee TD-HC becoming crucial as benchmarks catch up to the demands of truly long-horizon planning.
>
> ---
>
> > Is planning with the one-step GHM ($\gamma=0$) beneficial?
>
> We have directly tested this by re-running ActionPlan using our pre-trained GHM queried at $\gamma=0$ (i.e., as a one-step world model) in place of the dedicated one-step model.
>
> ||GHM ($\gamma=0$)|World Model|
> |:-|-:|-:|
> |antmaze-medium, gcbc|0.65|0.60|
> |antmaze-large, gcbc|0.35|0.34|
> |antmaze-giant, gcbc|0.01|0.00|
> |cube-1, gcbc|0.92|0.98|
> |cube-2, gcbc|0.40|0.37|
> |cube-3, gcbc|0.17|0.28|
> |cube-4, gcbc|0.00|0.03|
> |Average|0.36|0.37|
>
> As can be seen in the above table, the results are unchanged, suggesting that multi-horizon GHM training does not degrade one-step prediction quality, the GHM seamlessly serves as both a one-step and multi-step model. This further underscores the value of GHMs: the same model can match a dedicated one-step predictor and also enable compositional planning across a continuum of horizons.
>
> ---
>
> > Novelty concerns and limited downstream benefit of consistency loss
>
> The reviewer makes a fair observation but we would like to add some context: making accurate predictions at 200-500 step horizons is itself a notable achievement, as prior GHM methods exhibit systemic bias well before this regime. Figure 8 illustrates this, without consistency, samples traverse walls at long horizons. We believe this capability is a step change for the field. The reviewer is exactly right that current planning horizons don’t fully exercise TD-HC’s strengths, and this points to what we see as a broader gap: available benchmarks have not yet caught up to the horizon lengths that methods like ours can now handle. We see TD-HC as laying essential groundwork for the next generation of long-horizon planning problems, where prediction fidelity at these timescales will be the bottleneck.
>
> Moreover, although we report results using a U-Net GHM, we also experimented with transformer-based GHMs. In these experiments, training with TD-Flow consistently diverged in antmaze domains for long horizons, whereas TD-HC stabilized optimization. This suggests that TD-HC can significantly ease long-horizon GHM training, without requiring extensive tuning of optimization hyperparameters.

---

> > ### Author Rebuttal · Reviewer_QdoD · 2026-04-02
> >
> > Thank you for the additional experiments and clarifications. The distinction from GGPI is much clearer to me now, and I also appreciate the added short-horizon and γ=0 analyses. Please add it to the revised version.
> >
> > ### Remaining Concern
> >
> > While I think the paper already earned the positive score, I am a bit torn between holding my current score and increasing it. My remaining concern is about the causal link that seems to underlie the paper's broader motivation, namely:
> >
> > > better long-horizon world modeling ⟹ better evaluation of policy sequences ⟹ better planning performance
> >
> > The rebuttal now establishes that
> >
> > > TD-HC ⟹ better long-horizon prediction
> >
> > and also that the overall planning method performs well. However, it is still not fully clear to me that the key implication
> >
> > > better long-horizon prediction ⟹ better planning
> >
> > is demonstrated on the current benchmarks.
> >
> > At the moment, the evidence also seems consistent with an alternative explanation of the form
> >
> > > good short-horizon prediction + jumpy planning formulation ⟹ good planning
> >
> > In particular, since the one-step planning results also appear similarly strong (seen in the rebuttal), I am still unsure whether the main downstream benefit comes specifically from improved long-horizon modeling.
> >
> > ### What would help clarify this
> >
> > A brief side-by-side comparison of **one-step prediction** for **TD-Flow vs. TD-HC** is still missing from the rebuttal. Reporting this alongside the existing horizon-based results would help clarify whether the downstream planning benefit is specifically tied to improved long-horizon modeling or whether strong one-step prediction is sufficient. If this point is resolved more clearly, I would be more inclined to increase my score.
> >
> > Q1. Is TD-HC not only a better long horizon predictor but also a better 1-step predictor?
> >
> > Q2. Do you think the current case is: better long-horizon prediction ⟹ better short-horizon prediction + jumpy planning formulation ⟹ better planning?
> >
> > Q3. Given the close, strong results for both the one-step WM and the full WM, Do you intend to claim the causal link: better long-horizon prediction ⟹ better planning?
> >
> > I would appreciate this clarification and revision to the main text to show this to the readers.

---

> > > ### Author Response · Authors · 2026-04-05
> > >
> > > We sincerely thank the reviewer for their continued engagement and thoughtful follow-up. We appreciate the care with which they have analyzed the casual structure of our claims, and we hope the clarification below fully resolves the remaining concern.
> > >
> > > ## Clarifying the rebuttal table: GHM($\gamma = 0$) vs. World Model
> > >
> > > Upon reflection, we realize our earlier table was not presented clearly enough. In that table, both columns report ActionPlan results – the only difference is whether ActionPlan uses a dedicated one-step world model or our GHM queried at $\gamma = 0$. The purpose was to show that multi-horizon GHM training does not degrade one-step ($\gamma = 0$) prediction quality, i.e., the GHM can serve reliably as both a one-step and multi-step model.
> > >
> > > Crucially, neither column in that table corresponds to CompPlan. To illustrate the gap, we’ve amended that table with another column to include CompPlan:
> > >
> > > |||ActionPlan (GHM($\gamma = 0$)) |ActionPlan (Dedicated)|CompPlan|
> > > |:-|:-|-:|-:|-:|
> > > |gcbc|antmaze-giant|0.01|0.00|**0.03**|
> > > |gcbc|antmaze-large|0.35|0.34|**0.73**|
> > > |gcbc|antmaze-medium|0.65|0.60|**0.85**|
> > > |gcbc|cube-1|0.92|0.98|**0.99**|
> > > |gcbc|cube-2|0.40|0.37|**0.97**|
> > > |gcbc|cube-3|0.17|0.28|**0.92**|
> > > |gcbc|cube-4|0.00|0.03|**0.76**|
> > > ||Average|0.36|0.37|**0.75**|
> > >
> > > The contrast is stark. CompPlan achieves an average success of 75% – more than double of both ActionPlan variants. We apologize for any ambiguity in our rebuttal.
> > >
> > > ## The causal link: jumpy multi-horizon modeling ⟹ better planning
> > >
> > > We believe the casual link the reviewer is looking for is already established by the CompPlan vs. ActionPlan comparison.  Both methods use identical planning machinery --- random shooting, same number of candidates, same objective, same policies, same planning budget. The only difference is the temporal granularity of the world model. ActionPlan's one-step model is accurate at each individual step, but chaining these predictions over hundreds of steps degrades trajectory-level accuracy and requires searching over an exponentially large action space. GHMs sidestep both issues: by predicting directly over extended horizons, they give the planner access to reliable long-range predictions that one-step chaining cannot deliver in practice, while also shortening the effective planning depth from hundreds of steps to a handful of jumps. This is what drives CompPlan’s 201% relative improvement over ActionPlan on long-horizon tasks.
> > >
> > > TD-HC improves prediction quality within the jumpy planning framework, particularly at the longest horizons (rebuttal EMD results, Figure 8). On current benchmarks, planning queries moderate horizons where both TD-Flow and TD-HC are already sufficiently accurate, yielding a modest ~5% improvement (Table 9). The large gains come from having multi-horizon predictions at all, not from the incremental accuracy improvements that TD-HC provides over TD-Flow at short horizons. We anticipate TD-HC’s modeling advantage -- most pronounced at >200 step horizons -- will translate more directly into planning gains as benchmarks begin to demand reasoning at those timescales.
> > >
> > > ---
> > >
> > > Answering the reviewer’s questions directly:
> > >
> > > **Q1**: Both methods share the same one-step loss, so we would expect negligible difference. To confirm this directly, we ran the experiment the reviewer suggested – below we report one-step ($\gamma=0$) MSE for TD-Flow vs. TD-HC:
> > >
> > > |Domain|TD-Flow|TD-HC|
> > > |:-|:-|:-|
> > > |cube-1|0.06|0.06|
> > > |cube-2|0.12|0.12|
> > > |cube-3|0.16|0.16|
> > > |cube-4|0.19|0.19|
> > > |antmaze-giant|1.50|1.70|
> > > |antmaze-large|1.09|1.09|
> > > |antmaze-medium|1.08|1.06|
> > >
> > > As expected, the two methods are essentially identical at the one-step level across nearly all domains with antmaze-giant having the largest overall MSE. Overall, these results are consistent with our rebuttal EMD findings: the gap between TD-Flow and TD-HC is smallest at short horizons and grows with horizon length, confirming that TD-HC’s benefits are primarily at long horizons.
> > >
> > > **Q2**: The large planning gains come from multi-horizon jumpy modeling itself (CompPlan vs. ActionPlan). TD-HC provides orthogonal value by improving generative fidelity at the longest horizons. These are two complementary pieces rather than a sequential causal chain.
> > >
> > > **Q3**: To echo the above --- our claim is multi-horizon jumpy modeling ⟹ better planning, and we believe the CompPlan vs. ActionPlan comparison establishes this clearly. TD-HC addresses a complementary question about prediction fidelity within the jumpy framework, whose full downstream impact we expect to emerge as benchmarks grow in horizon length.
> > >
> > > ---
> > >
> > > We will revise the manuscript to better delineate how jumpy multi-horizon modeling drives planning performance and how TD-HC complements this with improved long-horizon prediction quality. This distinction was not sharp enough in the original submission, and we are grateful to the reviewer for pushing us to clarify it.

---

### Decision · Program_Chairs · 2026-04-30

**Decision:**

Accept (regular)

**Comment:**

The paper introduces a framework for compositional planning over pre-trained policies using jumpy world models trained as generative models of successor measures across multiple horizons. The method consists of two components: a TD-Flow model with a horizon-consistency objective that enforces coherent predictions across different jump lengths, and a planning formulation for evaluating policy sequences with variable stage-wise switching schedules. The approach integrates successor-measure modelling, temporal abstraction, and generative world models into a coherent theoretical formulation. Experimental results on OGBench demonstrate strong improvements on long-horizon navigation and manipulation tasks relative to several baselines.

Reviews were consistently positive about this work. The framework generalises several existing paradigms (generalised policy improvement, action-level planning, and geometric generalised policy improvement) into a single principled formulation, and the extension to variable switching probabilities over continuous policy families is a strong conceptual contribution. The empirical results are also strong, particularly on compositional manipulation tasks. The ablations are thorough and carefully isolate the contribution of each component. The presentation is clear, and the theoretical grounding of the planning estimator is sound. The rebuttal further strengthened the paper with simulated Humanoid results, confirming that the approach scales to higher-dimensional, longer-horizon settings.

The authors are encouraged to update the paper to include these new results and up[dated discussion points